# Divergent mechanisms of steroid inhibition in the human ρ1 GABA_A receptor

Chen Fan [1,2,3], John Cowgill [2,3], Rebecca J. Howard [1,2] ✉ & Erik Lindahl[1,2] ✉

ρ-type γ-aminobutyric acid-A (GABA_A) receptors are widely distributed in the retina and brain, and are potential drug targets for the treatment of visual, sleep and cognitive disorders. Endogenous neuroactive steroids including β-estradiol and pregnenolone sulfate negatively modulate the function of ρ1 GABA_A receptors, but their inhibitory mechanisms are not clear. By combining five cryo-EM structures with electrophysiology and molecular dynamics simulations, we characterize binding sites and negative modulation mechanisms of β-estradiol and pregnenolone sulfate at the human ρ1 GABA_A receptor. β-estradiol binds in a pocket at the interface between extracellular and transmembrane domains, apparently specific to the ρ subfamily, and disturbs allosteric conformational transitions linking GABA binding to pore opening. In contrast, pregnenolone sulfate binds inside the pore to block ion permeation, with a preference for activated structures. These results illuminate contrasting mechanisms of ρ1 inhibition by two different neuroactive steroids, with potential implications for subtype-specific gating and pharmacological design.

The neurotransmitter-gated γ-aminobutyric acid-A (GABA_A) receptors are anion-permeable pentameric ligand-gated ion channels expressed throughout the nervous system and other tissues. In response to binding the neurotransmitter GABA at an orthosteric site in the extracellular domain (ECD), a series of allosteric conformational changes open a pore over 50 Å away in the transmembrane domain (TMD), allowing anions (typically chloride) to transit the lipid bilayer[1]. In the continued presence of GABA, this activated open state typically transitions to a more thermodynamically stable desensitized state, with ion permeation occluded at the inner mouth of the TMD pore. Each subunit of the ECD contains 10 strands (β1–β10) interspersed by loops, some of which contribute to agonist binding; each subunit of the TMD contains 4 helices (M1–M4), with the M2 helices surrounding the central pore. In humans, GABA_A receptors are homo- or hetero-pentamers formed from a selection of 19 different subunits (α1-6, β1-3, γ1-3, ρ1-3, δ, ε, π and θ).

Although the ρ subtype is similar in sequence and structure to other GABA_A receptors, it was previously named GABA_C due to its distinct physiological and pharmacological properties[2]. These include insensitivity to bicuculline and sensitivity to the ρ-type specific inhibitor (1,2,5,6-tetrahydropyridin-4-yl)methylphosphinic acid (TPMPA). Of the three ρ GABA_A-receptor subtypes found in mammals, ρ1 is located predominantly in the retina; ρ2 is more widely distributed in the brain, including the cerebellum, thalamus, and frontal cortices; and ρ3 is found in the hippocampus and cerebellum[3–5]. These channels play roles during earlier postnatal neurodevelopment[5] and as potential therapeutic targets for post-stroke motor recovery[6]. There is increasing interest in developing drugs specific to ρ-type GABA_A receptors[7]. To better understand this system, we recently reported electron cryomicroscopy (cryo-EM) structures of the human ρ1 GABA_A receptor (henceforth termed ρ1) to 2.3 Å resolution in the absence and presence of classic agonists and inhibitors[8]. These structures were facilitated by deleting the flexible N-terminal region and intracellular M3-M4 loop from the wild-type sequence, generating the modified construct ρ1-EM. These modifications improve experimental accessibility while preserving wild-type functional features, enabling opportunities to characterize binding and modulation by pharmacologically relevant compounds.

Interestingly, a number of endogenous neuroactive steroids have been found to modulate GABA_A receptors, including ρ1[9]. A site for

[1]Dept. of Applied Physics, Science for Life Laboratory, KTH Royal Institute of Technology, Solna, Sweden. [2]Dept. of Biochemistry and Biophysics, Science for Life Laboratory, Stockholm University, Solna, Sweden. [3]These authors contributed equally: Chen Fan, John Cowgill. ✉e-mail: rebecca.howard@dbb.su.se; erik.lindahl@dbb.su.se

steroid potentiation at the transmembrane subunit interface, facing the inner membrane leaflet, has been described in some detail; notably, allopregnanolone, which is synthesized from progesterone locally in the brain, was recently resolved by cryo-EM at this site between β and α subunits in classical synaptic α1β2γ2 GABA$_A$ receptors[10,11]. The therapeutic relevance of such agents has received increasing attention with the effectiveness of allopregnanolone and its synthetic derivative, zuranolone, in treating post-partum depression[12]. In addition to positive modulators like allopregnanolone, several neuroactive steroids have been shown to inhibit the ρ1 subtype, although the mechanistic basis of negative modulation remains controversial[9]. Compounds that negatively modulate ρ1 include sulfated neurosteroids and β-estradiol (E2)[13].

Pregnenolone sulfate (PS) was one of the first identified neurosteroids, that is, steroids synthesized locally in the central or peripheral nervous system[14]. It is thought to exert excitatory effects, in part by suppressing neuro-inhibitory signaling via GABA$_A$ receptors[15]. The specific site(s) and mechanism of PS inhibition are unclear, though physiological, biochemical, and recent structural evidence support a role for pore-facing residues in classical synaptic GABA$_A$ receptors[4,10,16–18]. The estrogen steroid E2 is the major female sex hormone, involved in the development of the reproductive system and secondary sex characteristics, and in regulation of the menstrual cycle[19]. This hormone is mainly produced in ovaries, but also in other tissues including the brain, and is correlated with mood disorders[20]. It primarily binds and activates two nuclear receptors[21,22], but also mediates rapid and non-genomic effects via membrane proteins such as the G-protein coupled estrogen receptor[23]. Estrogens have also been shown to mediate rapid actions on ligand-gated ion channels, for example, potentiating human α4β2 neuronal nicotinic[24] and NMDA receptors[25]. In contrast, E2 effects on ρ1 are inhibitory, suggesting a notably distinct mechanism of modulation. Studies employing mutational analysis[13], voltage-clamp fluorometry[26], and functional modeling[27] have shown that E2 and sulfated steroids bind to distinct sites and act through different mechanisms, though their respective details remain to be characterized.

Here, by combining five cryo-EM structures with electrophysiology and molecular dynamics simulations, we characterize the binding sites and negative modulation mechanisms of E2 and PS at ρ1. We find that E2 binds in a pocket at the ECD-TMD interface, apparently specific to the ρ subtypes, and disrupts allosteric conformational changes linking GABA binding to pore opening. In contrast, PS binds inside the pore to block ion permeation, with a preference for activated structures. These results illuminate contrasting mechanisms of ρ1 inhibition by two different neuroactive steroids, with potential implications for subtype-specific gating and pharmacological design.

## Results

### E2 binds at the ECD-TMD interface of ρ1-EM

To explore distinctive steroid pharmacology in ρ1, we first characterized the functional effects of E2 (Fig. 1a) in our ρ1-EM construct. In agreement with previous reports[13], 30 μM E2 reduced ρ1-EM currents in *Xenopus* oocytes by roughly half in the presence of 1 μM GABA (~ EC$_{50}$) (Fig. 1b). Hypothesizing that E2 preferentially stabilizes a resting-like state of ρ1, we then determined a cryo-EM structure of ρ1-EM with E2. Like all structures in this and our previous work[8], the receptor was reconstituted in saposin nanodiscs with polar brain lipids. We identified a single predominant conformation to an overall resolution of 2.5 Å with C5 symmetry (Supplementary Figs. 1–3 and Table 1). Although E2 has a similar backbone to neurosteroids like allopregnanolone, the intersubunit transmembrane site previously shown to bind allopregnanolone in α1β2γ2[10] only contained tubular densities in the ρ1-EM/E2 complex, similar to those observed in apo ρ1-EM[8] and likely corresponding to phospholipid tails (Supplementary Fig. 3c). Instead, we observed a density corresponding in size and

shape to E2 at the ECD-TMD interface of each pair of adjacent subunits (Fig. 1c, d).

As verified by its protruding C16 methyl group (Fig. 1e), E2 fit unambiguously into this inter-domain density, with its C3 hydroxyl pointing up (toward the extracellular side), and C17 hydroxyl down (toward the intracellular side) (Fig. 1e, f). Overall, the E2 pocket is amphiphilic with local regions of positive charge (Supplementary Fig. 4a, b). The site is capped from the extracellular side by the β1-β2 loop and loop F, particularly the polar side chains of E113 and Q247 proximal to the C3 hydroxyl of E2 (Fig. 1e). From the transmembrane side, each E2 molecule is partially buried in a pocket enclosed by the upper M2-M3 region of the principal subunit, and by the upper M1 and M2 helices of the complementary subunit. On one face, the side chains of S334 and R337 are positioned to make hydrophobic and π-orbital interactions with E2 rings A and D, respectively (Fig. 1e, f). The opposite face approaches the hydrophobic surface of aromatic residues F283 and F284 at the amino-terminus of M1. Notably, substituting tyrosine for phenylalanine at these two positions largely ablated E2 inhibition while preserving GABA activation, indicating the precise geometry of this site critically determines E2 action (Fig. 1b, hand Supplementary Fig. 5b, c).

To our knowledge, small-molecule binding has not been previously shown for this pocket in any other GABA$_A$-receptor structure. In the presence of E2, ρ1-EM is nearly identical to the previously reported apo structure (Fig. 2f), indicating that the steroid does not induce substantial conformational change. Even the local configuration of the binding pocket is preserved, with side chain rotamers of the surrounding residues maintained relative to the apo structure (Fig. 1g). In contrast, GABA binding rearranges residues including R337 in this region[8], resulting in a pocket incompatible with E2 binding (Supplementary Fig. 4c). Interestingly, several residues proximal to E2, including F283, S334 and R347, were conserved among ρ1/2 but not α, β or γ subfamilies of GABA$_A$ receptors (Supplementary Fig. 6a). Moreover, no pocket capable of accommodating E2 was evident at any equivalent interface in the α1β2γ2 type (Supplementary Fig. 6b). In line with previous reports that classical synaptic GABA$_A$ receptors are insensitive to direct E2 modulation[28], these comparisons suggest a ρ-specific binding site and inhibitory mechanism, which could inform future pharmacological design.

To explore the specificity of this E2 site, we aligned the rings of several related steroids into the ρ1-EM/E2 complex. The inter-domain site appears to accommodate 17α-estradiol, while the enantiomer ent-17β-estradiol clashes with the side chain of M2 residue S334 (Supplementary Fig. 4d). Consistent with these models, 17α-estradiol was previously shown to inhibit ρ1 similar to E2, while ent-17β-estradiol lacks modulatory effect[13]. The 5α neurosteroids allopregnanolone and tetrahydrodeoxycorticosterone (THDOC) are among the most structurally similar to E2, yet they have both been shown to potentiate rather than inhibit ρ1[4]; the C3 hydroxyls of both these steroids are predicted to clash with R337 in our structures, suggesting they bind to an alternative site and/or state of the channel.

### E2 suppresses activating transitions of the ECD upon GABA binding

To further investigate the structural basis for E2 modulation, we also solved the structures of ρ1-EM in the presence of both E2 and GABA. Under these conditions, we identified two well-resolved classes in the same dataset (Supplementary Fig. 1 and Table 1). One class, comprising 58% of assigned particles, was largely superimposable with the previously reported GABA-bound structure, activated by five molecules of GABA and assigned to a desensitized state[8] (Fig. 2a, left). Notably, no E2 could be resolved in this structure. A second class, comprising 42% of assigned particles, also contained GABA in the orthosteric ligand sites, but with a global conformation markedly different from the desensitized state (Fig. 2a, right; Supplementary Fig. 3d, e). GABA binding in

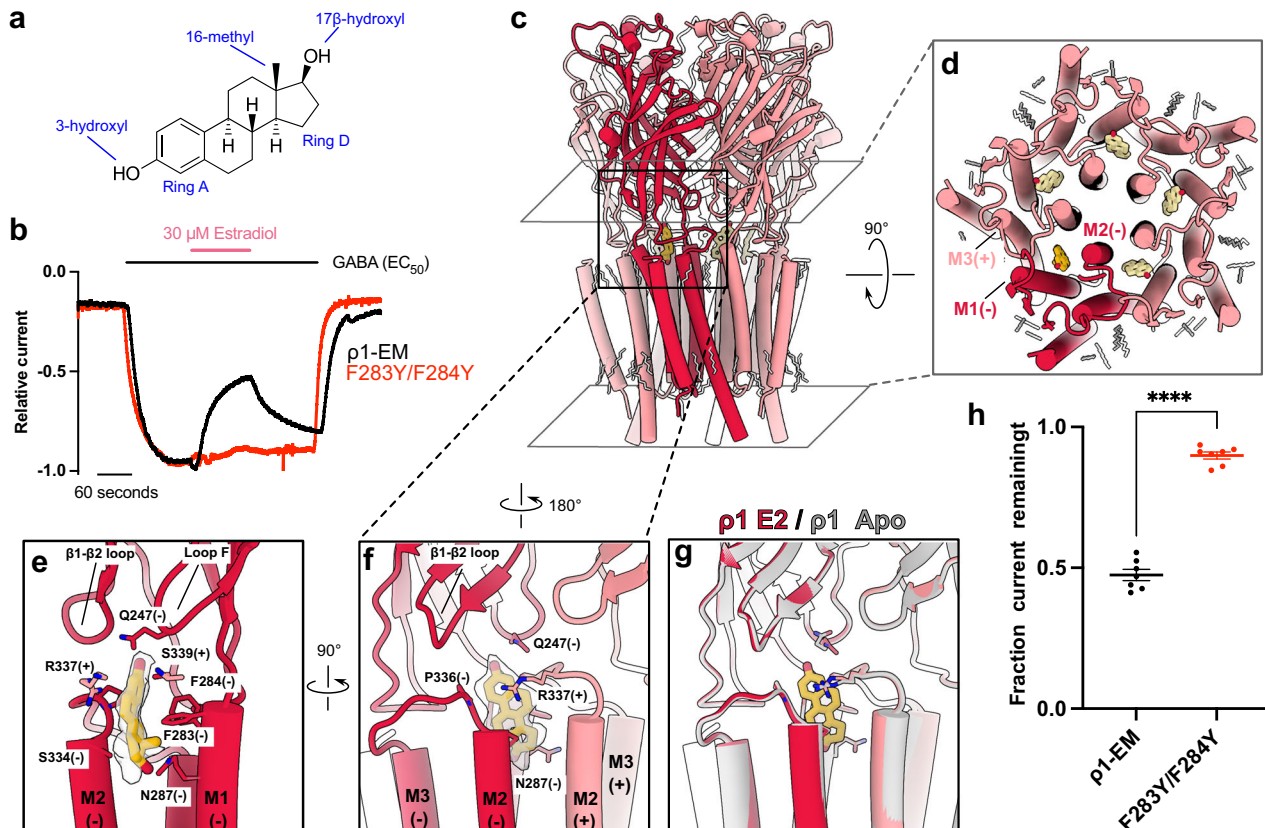

**Fig. 1 | E2 binds at the ECD-TMD interface of ρ1-EM. a** Chemical structure of β-estradiol (E2). **b** Sample traces from TEVC recordings of wild-type (black) and F283Y/F284Y (red) ρ1-EM constructs in response to GABA, with and without E2. **c** Cryo-EM structure of ρ1-EM with E2, viewed from the membrane plane. One subunit of the pentamer is colored dark red for definition. E2 (yellow) and resolved lipids (gray) are shown as thick and thin sticks, respectively. **d** TMD of ρ1-EM with E2, depicted as in panel c but viewed from the extracellular side. **e** Zoom view of a single E2 binding site, viewed from the membrane plane relative to the complementary (−) face of a single ρ1-EM subunit. The density assigned to E2 is shown in

transparency. E2 and surrounding residues are shown as sticks and labeled. **f** Zoom view of a single E2 binding site, depicted as in panel e but rotated 90° to show the interface between two subunits from the channel pore. **g** Superimposed structures of apo (gray, PDB ID: 8OQ6) and E2-bound (red) ρ1-EM, showing no major change upon E2 binding. **h** Fractional current remaining after treatment with 30 μM E2 with ~ EC$_{50}$ GABA (1 μM for wild-type, 4 μM for F283Y/F284Y ρ1-EM constructs). Error bars represent SEM from 7 individual oocytes, and stars represent $p < 0.0001$ ($p = 6.016e-9$) from a two-way $t$ test.

this second structure was associated with only a minor ECD twist of 1.2° compared with apo or E2-only conditions (Supplementary Fig. 4e), 7.3° less than in the desensitized state (Fig. 2d), and the pore is closed (Fig. 2g). Accordingly, we assigned this structure to a liganded preopen state, possibly corresponding to one of the so-called primed states described in other pentameric ligand-gated ion channels[29,30]. We observed E2 in a site comparable to the E2-only complex (Fig. 2c), suggesting E2 disturbs allosteric GABA activation by wedging into the ECD-TMD interface between each pair of subunits.

The state dependence of E2 binding is reminiscent of the selective stabilization of picrotoxin (PTX) in the closed pore of ρ1-EM[8], in line with previous reports that these inhibitors act through related mechanisms[13]. Indeed, apparent GABA affinity was reduced in the presence of E2, consistent with stabilization of a resting-like state (Fig. 2e). Moreover, fractional E2 inhibition decreased with increasing concentrations of GABA (Supplementary Fig. 5a, c), precluding a purely noncompetitive mechanism (e.g., pore block). A modest apparent reduction in maximal GABA efficacy (Fig. 2e) may represent an artifact of slow desensitization contributing to the steady-state inhibited current; indeed, this effect persisted at all E2 concentrations in both wild-type and F283Y/F284Y constructs (Supplementary Fig. 5a, c). On the other hand, given the limited effect of E2 with saturating GABA in electrophysiology experiments, it may seem surprising that E2 promotes such a substantial population in the primed state by cryo-EM, even under high-GABA conditions that would

produce a single desensitized state without E2[8]. However, concentrations of E2 applied in the cryo-EM samples are roughly an order of magnitude higher than concentrations used in electrophysiology, due to the improved solubility of E2 in the presence of lipids and detergents used in grid preparation. Thus, it is difficult to directly assess the functional effect of E2 at cryo-EM concentrations. Interestingly, an overlay of the E2- and PTX-bound structures in the presence of GABA shows that domain twist is even more limited by E2 than by PTX (Supplementary Fig. 4f), in line with previous voltage-clamp fluorometry data showing that the steroid suppresses upper-ECD rearrangements more than the toxin[26].

## PS occludes the GABA-bound state

To explore alternative mechanisms of ρ1 inhibition, we then characterized the 3β-sulfated neurosteroid PS (Fig. 3a), a negative modulator of this and several other GABA$_A$-receptor subtypes[15,31]. Similar to previous reports[13], 100 μM PS reduced ρ1-EM currents by roughly 35%, though there was a small recovery of the inhibited current in the continued presence of PS (Fig. 3b). This partial reversal of inhibition during PS exposure has also been observed in classical synaptic GABA$_A$ receptors, attributed to a redistribution amongst functional states upon PS binding[31]. Unlike E2, PS wash-out was associated with a transient recovery current ~ 30% larger than steady-state GABA activation prior to treatment (Fig. 3b). This behavior is expected for a pore blocker that preferentially binds to the activated (open or

**Table 1 | Cryo-EM data collection, refinement, and validation statistics**

|  | E2 (EMD-19167) (PDB 8RH4) | GABA + E2 Primed (EMD-19171) (PDB 8RH7) | GABA + E2 Desensitized (EMD-19172) (PDB 8RH8) | PS (EMD-19173) (PDB 8RH9) | GABA + PS (EMD-19175) (PDB 8RHG) |
|---|---|---|---|---|---|
| **Data collection and processing** | | | | | |
| Magnification | 130,000 | 130,000 | 130,000 | 130,000 | 130,000 |
| Voltage (kV) | 300 | 300 | 300 | 300 | 300 |
| Electron exposure (e⁻/Å2) | 41.58 | 41.58 | 41.58 | 44.61 | 44.38 |
| Defocus range (μm) | − 0.8 to −1.8 | − 0.8 to −1.8 | − 0.8 to −1.8 | − 0.8 to − 1.8 | − 0.8 to − 1.8 |
| Pixel size (Å) | 0.6725 | 0.6725 | 0.6725 | 0.6725 | 0.6725 |
| Symmetry imposed | C5 | C5 | C5 | C1 | C1 |
| Final particles | 134,816 | 100,833 | 140,148 | 93,154 | 88,968 |
| Map resolution (Å) FSC threshold | 2.52 0.143 | 2.78 0.143 | 2.66 0.143 | 3.21 0.143 | 3.01 0.143 |
| Map resolution range (Å) | 2.3–2.7 | 2.6–3.0 | 2.5–2.9 | 3.0–3.8 | 2.9–3.7 |
| **Refinement** | | | | | |
| Initial model (PDB code) | 8OQ6 | 8OQ6 | 8OP9 | 8OQ6 | 8OP9 |
| Model resolution (Å) FSC threshold | 2.7 0.5 | 2.9 0.5 | 2.8 0.5 | 3.3 0.5 | 3.3 0.5 |
| Map sharpening $B$ factor (Å²) | − 97.2 | −112.2 | −109.6 | −101.0 | −97.9 |
| Model composition | | | | | |
| Non-hydrogen atoms | 14292 | 13866 | 14067 | 14247 | 13897 |
| Protein residues | 1660 | 1610 | 1645 | 1665 | 1650 |
| Ligands | 62 | 61 | 42 | 51 | 26 |
| $B$ factors (Å²) | | | | | |
| Protein | 25.67 | 34.71 | 41.36 | 88.77 | 94.92 |
| Ligand | 46.63 | 44.83 | 72.40 | 85.43 | 75.75 |
| R.m.s. deviations | | | | | |
| Bond lengths (Å) | 0.010 | 0.005 | 0.007 | 0.003 | 0.003 |
| Bond angles (°) | 1.261 | 1.105 | 1.265 | 0.628 | 0.501 |
| Validation | | | | | |
| MolProbity score | 1.10 | 1.47 | 0.94 | 1.38 | 1.28 |
| Clashscore | 2.05 | 3.48 | 1.58 | 4.71 | 5.10 |
| Poor rotamers (%) | 0.32 | 0.34 | 0.33 | 0.06 | 0.07 |
| Ramachandran plot | | | | | |
| Favored (%) | 97.38 | 95.28 | 97.85 | 97.26 | 97.98 |
| Allowed (%) | 2.62 | 4.72 | 2.15 | 2.74 | 2.02 |
| Disallowed (%) | 0.00 | 0.00 | 0.00 | 0.00 | 0.00 |

desensitized) state of the pore. Consistent with this model, current recovery upon PS washout was not observed at high concentrations of GABA, where channels are fully activated prior to PS treatment (Supplementary Fig. 7f).

To test this pore-block hypothesis, we determined a cryo-EM structure of ρ1-EM in the presence of PS and GABA. To avoid artifacts in a potential pore site, we processed these data without imposing symmetry, resolving a single predominant conformation to 3.2 Å (Supplementary Fig. 1 and Table 1). In addition to five GABA molecules at the orthosteric ECD sites, the resulting map contained a single density capable of accommodating PS, spanning residues P311 (− 2′) to L322 (9′) in the inner half of the channel pore (Fig. 3c, d and Supplementary Fig. 3d, e). At the inward-facing end of PS proximal to the − 2′ side chains, an additional spherical density was modeled as a chloride ion, also observed in our previous structures of ρ1-EM with GABA[8] (Fig. 3f).

The PS density could accommodate modeling in two possible poses, with the sulfate group either facing up (toward the 9′ activation gate) or down (toward the −2′ desensitization gate) (Supplementary Fig. 7a). We tested the orientation of PS by running four replicate

> 300 ns all-atom molecular dynamics (MD) simulations in each of the two poses (Table 2). Whereas the sulfate-up pose was relatively stable, the sulfate-down pose varied widely, displacing over a 14-Å range up or down the pore axis and > 5 Å median root-mean-square deviation (RMSD) (Supplementary Fig. 7b, c). Accordingly, we modeled PS with the sulfate up for all further analyses. We observed no other steroidal densities in the PS dataset, including in the E2 site or synaptic-subtype allopregnanolone site[10].

The PS pose in ρ1-EM overlapped that in a recently reported complex with the synaptic α1β2γ2 subtype, including the orientation of the sulfate group[10] (Fig. 3g). A pore-block mechanism has similarly been proposed in this synaptic subtype, supported by mutations in the inner pore that suppress inhibition[16] and disrupt PS stability in MD simulations[10]. The lower reported sensitivity of ρ1 versus synaptic subtypes to PS inhibition[17] may be attributable to sequence differences in the channel pore, particularly at 2′, which is occupied by proline in ρ1, valine in α1, alanine in β2, and serine in γ2 (Supplementary Fig. 6a). Indeed, substituting the equivalent α1 residue at 2′ in ρ1 (P315V) has been shown to increase PS sensitivity[17]. As previously reported[8], the ρ1 pore is also expanded relative to classical synaptic GABA_A-receptor

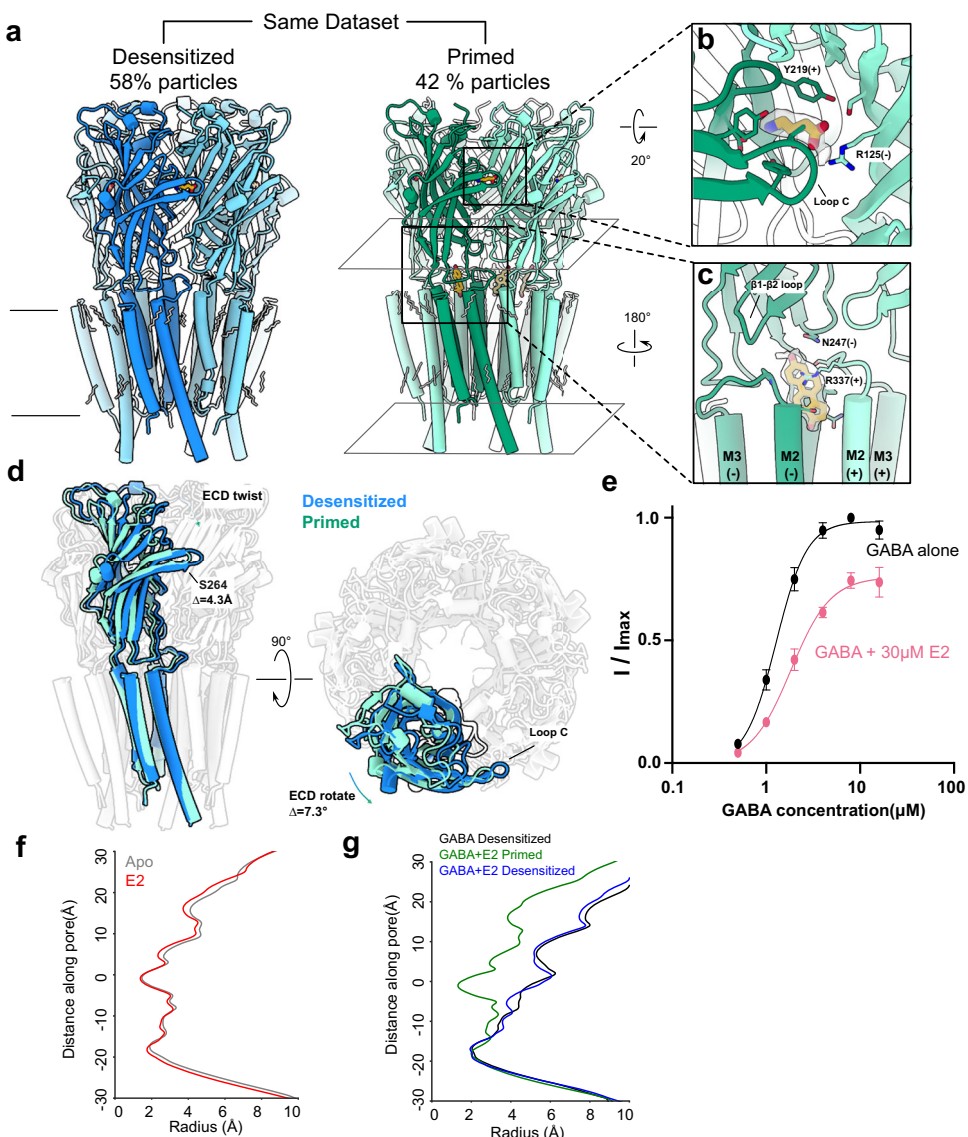

**Fig. 2 | E2 suppresses activating transitions of the ECD upon GABA binding.**
**a** Two different cryo-EM structures were obtained from a single sample of ρ1-EM with GABA and E2, viewed from the membrane plane. Presumed functional state (left, blue: desensitized; right, green: primed) and ensemble contribution as a percent of resolved particles are indicated above each structure. One subunit of each pentamer is colored darker for definition. E2 (yellow) and resolved lipids (gray) are shown as thick and thin sticks, respectively. **b** Zoom view of a single GABA binding site in the primed state, depicted as in panel a. Density assigned to GABA is shown in transparency. GABA and surrounding residues are shown as sticks and labeled. **c** Zoom view of a single E2 binding site in the primed state, depicted as in Fig. 1f from the channel pore. The density assigned to E2 is shown in transparency. E2 and surrounding residues are shown as sticks and labeled. **d** Superimposed structures of ρ1-EM in the apparent desensitized (blue) and primed (green) states, viewed from the membrane plane (left) and extracellular side (right). All but one subunit of each pentamer is rendered transparent for clarity. **e** GABA concentration-response curves for ρ1-EM in the absence (black) and presence of 30 μM E2 (pink). Error bars represent SEM from 5 individual oocytes. Solid lines represent fits Boltzmann curves with an $EC_{50}$ of 1.3 μM (GABA alone, 95% confidence interval 1.16–1.41 μM) or 1.8 μM (GABA + E2, 95% confidence interval 1.56–2.21 μM). **f** Pore-radius profiles of ρ1-EM apo (gray) and E2 (red) structures. Both structures are assigned to a resting-like state. **g** Pore-radius profiles of ρ1-EM in the presence of GABA alone (black, PDB ID: 8OP9) and GABA + E2 in the primed (green) and desensitized (blue) states.

structures in the desensitized state (Fig. 3g), which could weaken contacts with a pore-bound ligand like PS.

To further validate this blocking mechanism, we compared ρ1-EM functional inhibition by PS to other inhibitors. PS inhibition was more efficacious at more positive potentials (Fig. 3h and Supplementary Fig. 7h), as expected for a negatively charged blocker. In contrast, ρ1-EM inhibition by the neutral blocker PTX was largely independent of voltage (Supplementary Fig. 7h–j). We estimated the fraction of the electric field traversed by the charged moiety of PS upon block by fitting the voltage-dependence of inhibition to the Woodhull model[32]. This model relies on two free parameters, the fraction of the electric field traversed (δ) and the affinity of the blocker to the pore at 0 mV ($K_D^{0mV}$). Our data for the

PS block in ρ1-EM fit well to the Woodhull model with δ = 0.705 and $K_D^{0mV}$ = 36.3 μM, indicating the charge traverses ~71% of the electric field of the pore. Some increase in the apparent block at positive potentials may actually reflect increased channel activity with increasing voltage that also favors PS block, resulting in a potential overestimate in δ[33]. Still, this error is likely to be small given that PTX, which shows an inverted activity-dependent block relative to PS, is nearly voltage-independent (Supplementary Fig. 7h–j). Comparison to PTX profiles suggests the activity dependence of the block contributes 0.05–0.09 of the apparent δ value.

The binding of PS in the inner ρ1 pore was reminiscent of our previously reported complex with PTX and GABA, including contacts

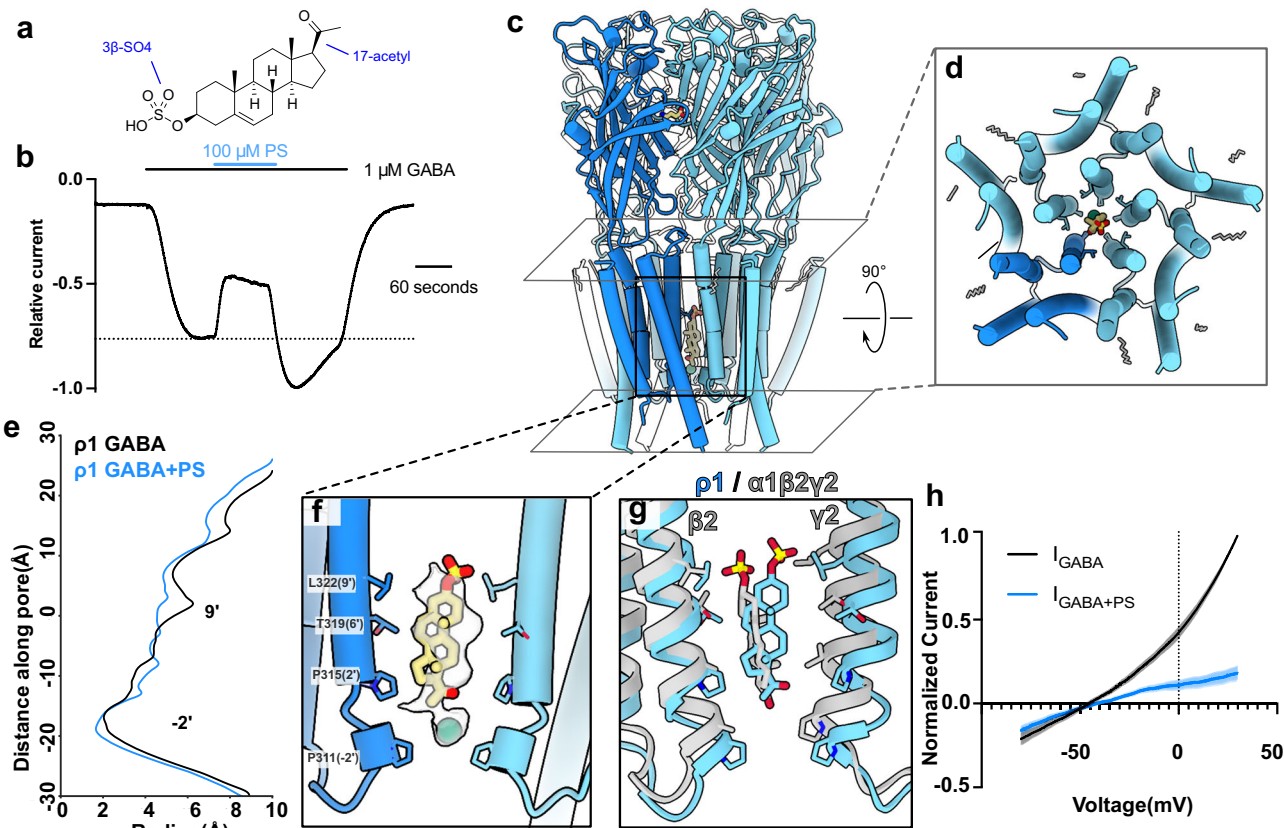

**Fig. 3 | PS occludes the GABA-bound state. a** Chemical structure of pregnenolone sulfate (PS). **b** Sample trace from TEVC recording of ρ1-EM in response to GABA and PS. The dotted line indicates maximum GABA response prior to PS application to highlight increased current upon washout of PS. **c** Cryo-EM structure of ρ1 with GABA and PS, viewed from the membrane plane. One subunit of the pentamer is colored dark blue for definition. PS (yellow) and resolved lipids (gray) are shown as thick and thin sticks, respectively. **d** TMD of ρ1-EM with GABA and PS, depicted as in panel c but viewed from the extracellular side. **e** Pore-radius profiles of ρ1-EM with GABA alone (black, PDB ID: 8OP9) and GABA + PS (blue). **f** Zoom view of the PS binding site. The density assigned to PS is shown in transparency. PS and inner pore-lining residues are shown as sticks and labeled. **g** Superimposition of structures with GABA and PS of ρ1-EM (blue) and α1β2γ2 (gray, PDB ID: 8SGO) GABA$_A$ receptors. **h** Background-subtracted and normalized current-voltage curves for voltage ramps applied to ρ1-EM in the presence of GABA alone (black) or in combination with 100 μM PS (blue). Shaded regions represent SEM from 4 individual oocytes.

at 2'. However, PTX selectively stabilizes an intermediate state in which the TMD is locked closed[8]. In contrast, the complex with PS and GABA was largely superimposable with our previous GABA-only structure, with an all-atom RMSD < 0.8 Å (Fig. 3e). Accordingly, the structure was presumed to be activated, occupying a desensitized state[8]. Modest changes were observed at either end of the PS site, subtly shifting the − 2' and 9' side chains towards the intracellular side and pore axis respectively (Supplementary Fig. 7d). Like PTX, PS decreased maximal GABA efficacy (Supplementary Fig. 7f, g); however, other electrophysiological properties distinguished the mechanisms of these two pore blockers. Whereas apparent GABA affinity decreases with PTX[8], it increases with PS (Supplementary Fig. 7g), consistent with the steroid stabilizing an activated- rather than resting-like state. In contrast to E2, fractional inhibition by PS increased with increasing concentrations of GABA (Supplementary Fig. 5c), again consistent with preferential binding upon channel activation. Along with the recovery current observed after PS washout (Fig. 3b), these functional properties support a distinctive mechanism of PS inhibition by entering and binding to stabilize the activated pore.

## PS has limited access to the resting-like pore

Finally, we determined a cryo-EM structure of ρ1-EM with PS alone, resolving a single conformation to 3 Å without imposing symmetry (Supplementary Fig. 1 and Table 1). As in the presence of GABA, we observed a PS-like density inside the pore, between the − 2' and 9'

positions (Fig. 4a, b and Supplementary Fig. 3d, e). PS in this structure was more stable with its sulfate group down rather than up in MD simulations, likely due to the 9' constriction precluding sulfate occupancy (Fig. 4c, d and Table 2). The steroid was also displaced 1.6 Å down toward the − 2' gate, compared to its center of mass in the presence of GABA (Fig. 4e). No other densities in this structure were consistent with PS binding.

The ρ1-EM PS complex was assigned to a resting-like state, with no ligand in the orthosteric ECD sites and a radius < 2 Å at both the − 2' and 9' gates (Fig. 4f). It was largely similar to the previously reported apo structure of ρ1-EM[8], with an all-atom RMSD of 0.3 Å. The most prominent difference was a modest expansion at the 2' and 6' positions, presumably to accommodate the proximal steroid rings of PS (Fig. 4e, f). PS occupancy in a closed pore was surprising, as the steroid radius is at least 5 Å, too large to transit the constrictions at either − 2' or 9'. Indeed, the bulky steroid rings never fully exited either the − 2' or 9' gates in our MD simulations. Using enhanced sampling simulations, we calculated a free-energy barrier > 50 kJ/mol for PS to pass the − 2' gate, and approaching 100 kJ/mol to pass 9' (Fig. 4g and Supplementary Fig. 8). The barrier at 9' was absent in the complex with PS and GABA (Fig. 4g), indicating the steroid can freely enter the pore from the extracellular side upon channel activation. Binding in the pore site was favorable relative to bulk, both in the PS structure and to an even greater extent in the structure with GABA + PS; this profile is consistent with occupancy in both cryo-EM

**Table 2 | System setup of MD simulations**

|  | PS only sulfate up | PS only sulfate down | GABA + PS sulfate up | GABA + PS sulfate down |
|---|---|---|---|---|
| Simulation box | 118 Å x 118 Å x 188 Å | 118 Å x 118 Å x 188 Å | 120 Å x 120 Å x 183 Å | 120 Å x 120 Å x 183 Å |
| Number of atoms | 269496 | 269496 | 268076 | 268076 |
| Number of waters | 62349 | 62349 | 61958 | 61958 |
| Salt concentration | 150 mM NaCl | 150 mM NaCl | 150 mM NaCl | 150 mM NaCl |
| Number of lipids | 186 POPC | 186 POPC | 186 POPC | 186 POPC |
|  | 124 POPE | 124 POPE | 124 POPE | 124 POPE |
|  | 26 POPS | 26 POPS | 26 POPS | 26 POPS |
|  | 70 Cholesterol | 70 Cholesterol | 70 Cholesterol | 70 Cholesterol |
|  | 20 Sphingomyelins | 20 Sphingomyelins | 20 Sphingomyelins | 20 Sphingomyelins |
|  | 18 PIP2 | 18 PIP2 | 18 PIP2 | 18 PIP2 |

structures and with preferential binding following activation of the 9' gate.

The structure with PS suggests that transient rearrangements in the course of cryo-EM sample preparation, on the timescale of more than 30 min ligand incubation, allow the steroid to bind in the resting-like state, with apparent alterations in orientation and pose to accommodate the constricted pore. However, structures in the presence of GABA show that a closed pore is not preferentially stabilized by PS as it is by PTX or E2, and the complex with PS and GABA appears to be the more relevant model for ρ1 functional inhibition.

## Discussion

Our structural, functional, and computational results reveal distinct sites of action and divergent inhibitory mechanisms for the neuroactive steroids E2 and PS on a ρ1 GABA$_A$ receptor. E2 binds at the ECD-TMD interface and appears to act as a wedge, blocking allosteric domain rearrangements that link ECD GABA binding to TMD pore opening (Fig. 5a). The absence of clear E2 density in the desensitized state of the GABA + E2 dataset suggests that full activation of the receptor precludes E2 binding. In contrast, the opening of the 9' activation gate enables PS to bind inside the pore, blocking ion permeation (Fig. 5b). Preferential stabilization of the activated receptor is clear from the increased apparent GABA affinity and transient increase in current amplitude upon PS washout at low GABA concentrations. No binding was evident for either agent at the inner-leaflet site classically associated with allopregnanolone potentiation of α1β2γ2 GABA$_A$ receptors; indeed, this site diverges in sequence especially at a key M1 position (α1-Q242/ρ1-W300), likely accounting for the limited allopregnanolone sensitivity of ρ1[9]. Although our results cannot entirely exclude transient occupancy of additional sites, they highlight the capacity of different steroids to modulate GABA$_A$ receptors via structurally distinct, largely exclusive mechanisms. Whereas the mechanism of PS inhibition we report here for ρ1 largely resembles that proposed for the classical synaptic α1β2γ2 GABA$_A$ receptor[10], the site of E2 inhibition appears specific to ρ subtypes.

The buried domain-interface site observed here for E2 is relatively unexplored as a direct mediator of allosteric modulation and, to our knowledge has yet to be visualized in any known pentameric ligand-gated ion channel structure. It is notably distant from steroid sites in previous GABA$_A$-receptor structures, potentially accessible from the extracellular medium rather than upon partitioning into the membrane core. Particularly in the absence of a definitive open state, the limited timescales of classical MD preclude definitive modeling of mechanistic effects of an allosteric ligand like E2; nonetheless, conformational changes in the desensitized state due to expansion of the upper pore are likely to also reflect the open state, and could similarly block E2 binding. The discovery of this evident modulatory site in ρ1 is particularly notable, given that this subfamily is thought to lack classical allosteric sites for benzodiazepines, barbiturates, and general

anesthetics. Development of ρ-specific modulators has focused largely on the orthosteric GABA site, where TPMPA and related compounds bind; the E2 site could constitute a novel development target. E2 itself is known to promote excitability in the hippocampus by suppressing GABA signaling, although this effect has been primarily attributed to the alteration of classical synaptic GABA$_A$-receptor expression via nuclear receptors[34]. Given its IC$_{50}$ (6.5 μM[13]) is nearly one hundred times higher than circulating levels (≤150 nM in non-pregnant women[35]), E2 inhibition of ρ1 may play a limited physiological role. Furthermore, therapeutic applications of this site would require selecting against other E2 targets. Nonetheless, this steroid appears to constitute a promising lead compound for the design of ρ-type specific inhibitors, potentially useful in the treatment of visual, sleep, or cognitive disorders[7].

Although long suspected, the pore block of GABA$_A$ receptors by sulfated neurosteroids has also been controversial, due in part to inconsistent evidence for voltage dependence in synaptic subtypes. Here, we demonstrate that the PS block of ρ1 is indeed mildly voltage-dependent, as expected from the negatively charged sulfate group interacting with the electric field across the pore. The expanded pore of ρ1 versus classical synaptic GABA$_A$ receptors in the presence of GABA[8] could contribute to the relative robustness of PS block to pore mutations[13]. Interestingly, inhibition by the related compound pregnanolone sulfate was previously shown to be voltage-dependent at α1β2γ2 GABA$_A$ receptors but voltage-independent at ρ1[13,33]. It is possible that PS and pregnanolone sulfate act at different sites; indeed, at least three distinct mechanisms of ρ1 inhibition have been proposed for different steroids[13]. Alternatively, subtle differences in the position of the sulfate group or local pore structure may position the charged moiety outside the electric field gradient. Although the structural details conferring voltage sensitivity (or lack thereof) on steroid inhibition remain unclear, our structures of PS-bound ρ1 combined with its electrophysiological profile, as well as molecular simulation data coherently support a pore-blocking mechanism for this agent.

Another interesting feature of PS inhibition is its enhancement by GABA activation. Unlike most steroids that primarily modulate ρ1 receptor function at low GABA concentrations[13], PS inhibits maximally at saturating GABA. This profile indicates that PS binds preferentially in the context of GABA activation, giving rise in cryo-EM to a partially open or desensitized state. Although PS binding in the resting-like state is evidently possible in the context of prolonged incubation for cryo-EM, it would be disfavored by the permeation barrier at the 9' gate. This barrier was estimated at 100 kJ/mol by our PMF calculations: although these measurements do not directly report conductance, the central hydrophobic gate is expected to preclude inward transit of PS as well as chloride ions in a physiological setting. Furthermore, the recovery current apparent upon PS washout supports preferential binding to an open rather than desensitized state.

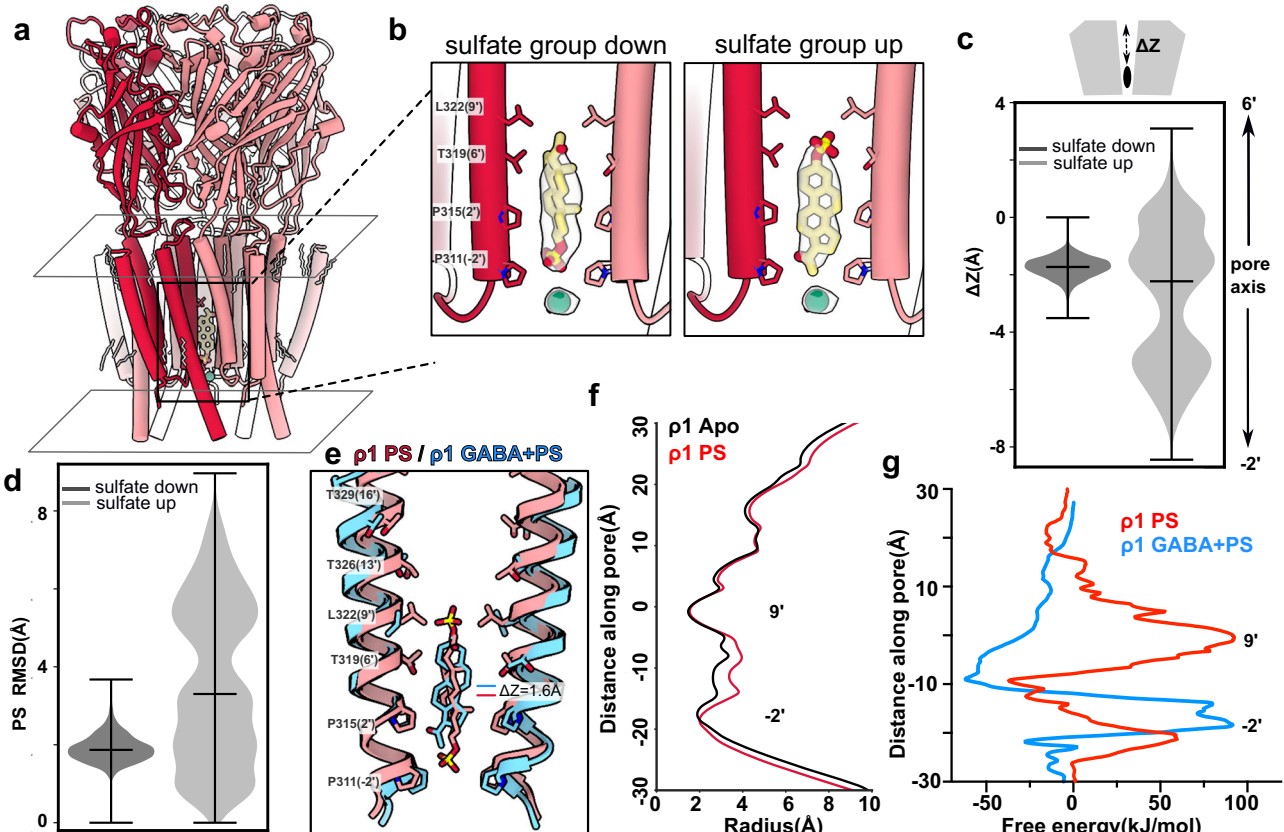

**Fig. 4 | PS has limited access to the resting-like pore. a** Cryo-EM structure of ρ1-EM with PS, viewed from the membrane plane. One subunit of the pentamer is colored dark red for definition. PS (yellow) and resolved lipids (gray) are shown as thick and thin sticks, respectively. **b** Zoom views of the inner pore of ρ1-EM with PS, with experimental density assigned to PS and chloride shown in transparency. Two possible poses are shown for PS, either with the sulfate group oriented down towards the cytosol (left) or up towards the 9′ hydrophobic gate (right). PS (yellow), chloride (green), and surrounding residues are shown as sticks and labeled. **c** Translocation of PS along the pore z-axis in MD simulations launched from the two poses shown in B. Simulation frames are aligned on the Cα atoms of the M2 pore-lining helices, and translocation (ΔZ) calculated for the center of mass of PS non-hydrogen atoms along a linear axis passing through the channel pore. Violin

plots represent probability densities from 4 independent simulation replicates of > 300 ns each, sampled every 0.4 ns ($n$ = 3019 and 3271 for simulations with the sulfate oriented down and up, respectively). In each plot, the middle marker indicates the median, and error bars indicate minimum and maximum values. **d** Mobility of PS in MD simulations, calculated from RMSD of PS non-hydrogen atoms, with sample sizes and plot parameters as in panel (**c**). **e** Superimposition of the structures of ρ1-EM with PS alone (red) and with GABA + PS (blue). PS and pore-lining residues are shown as sticks and labeled. **f** Pore-radius profiles of apo (black, PDB ID: 8OQ6) and PS-bound (red) structures of ρ1-EM. **g** Potential of mean force-free energy for PS movement along the pore axis in PS-bound structures of ρ1-EM in the presence (red) and absence (blue) of GABA (9′ gate at 0 nm).

On the other hand, minor rearrangements are apparent in our structures with GABA versus GABA + PS, including a modest constriction of the inner desensitization gate relative to the structure with GABA alone. In contrast, our past and current work with inhibitors favoring the resting-like state of ρ1 (TPMPA, PTX, E2) demonstrates these compounds require little or no rearrangement around the respective binding sites, even on a local scale. These findings may indicate that cryo-EM sample conditions favor a desensitized-like structure that does not perfectly represent the predominant physiological ensemble[36], which should include one or more open structures. Alternatively, these results may support a recent hypothesis based on detailed kinetic modeling of the α1β3γ2 GABA$_A$ receptor, where PS binding stabilizes a nonconducting state distinct from both open and desensitized[31]. Such a model would recapitulate several functional features we observe here, including increased apparent GABA affinity in the presence of PS and a transient increase in current amplitude upon PS washout.

Taken together, our findings expand on a growing body of literature demonstrating that despite similar structural backbones, neuroactive steroids can have diverse binding sites and mechanisms of action on GABA$_A$ receptors. The importance of neuroactive steroids as building blocks for new therapies is clear, given the recent success of

the endogenous modulator allopregnanolone and synthetic derivative zuranolone in the treatment of postpartum depression[12]. The structures we report here can aid future structure-based drug design to better target ρ-type receptors, which are insensitive to nearly all classical GABA$_A$ receptor-targeting therapies.

## Methods

### Protein expression and purification

The expression-optimized human ρ1 construct (ρ1-EM) was expressed and purified according to previous methods[8]. Briefly, baculovirus encoding ρ1-EM was amplified by infecting a 300-mL suspension Sf9 cell culture (Novagen). Expi293F cells (Gibco) were infected by baculovirus at a density of $2 \times 10^6$ cells/mL. After 6 h incubation at 37 °C, 5 mM sodium butyrate was added and the cells were further cultured at 30 °C for 48 h. Cells were harvested, washed with phosphate-buffered saline, and then flash-frozen until further usage.

For sample preparation of the PS datasets, cell pellets from 2 L culture were resuspended in resuspension buffer (40 mM HEPES pH 7.5, 300 mM NaCl, with cOmplete protease inhibitor tablets (Roche)) and sonicated to break cell membranes. The membranes were pelleted by ultracentrifugation, then resuspended and solubilized in resuspension buffer with 2% lauryl maltose neopentyl glycol (LMNG) and

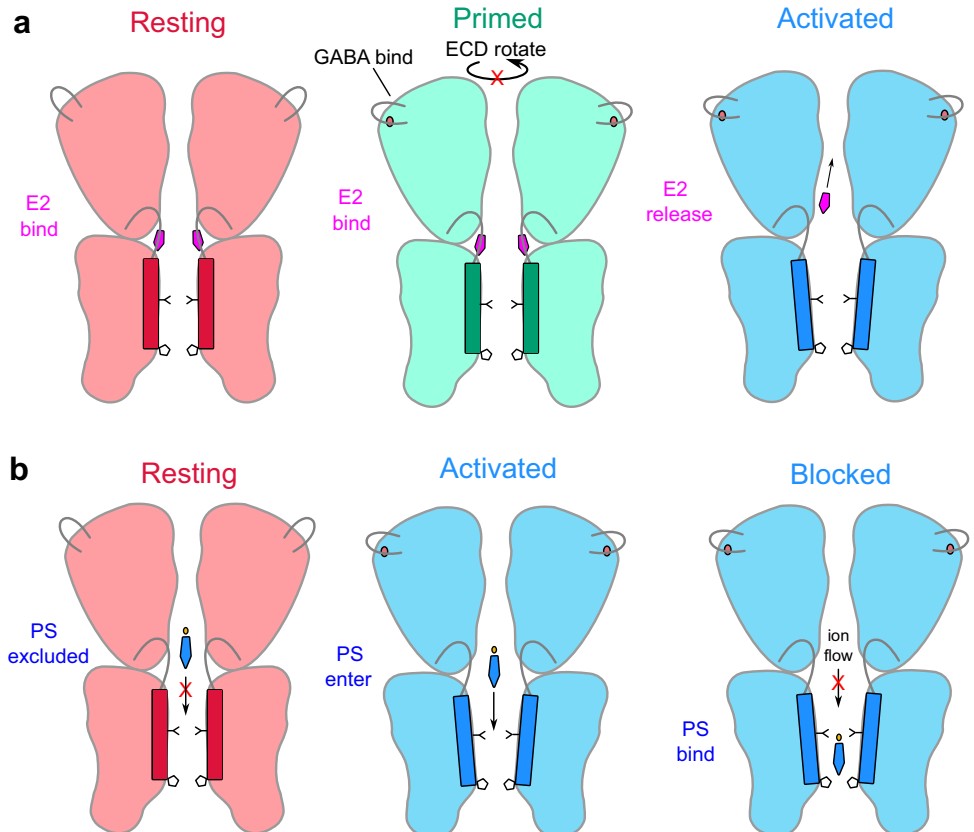

**Fig. 5 | Negative modulation mechanisms of ρ-type GABA$_A$ receptors by steroids evidenced in this work.** Cartoons are derived from structures of ρ1-EM determined in this and previous work. Structures without GABA (red) in the absence or presence of E2 correspond to resting-like states. One structure with GABA and E2 (green) is assigned to a primed state. Otherwise, activation by GABA (blue) induces agonist-induced transitions in the ECD and 9' hydrophobic gate, which are retained in the context of the PS block; corresponding experimental structures are parsimoniously assigned to desensitized states. **a** E2 wedges into the ECD-TMD interface to disrupt allosteric conformational transitions linking GABA binding to ECD rotation and pore opening. **b** PS binds inside the pore to block ion permeation, with an apparent preference for activated structures.

0.2% cholesteryl hemisuccinate (CHS) for 3 h in a cold room (4–10 °C). The solubilization mixture was ultracentrifuged and the supernatant was applied to 4-mL Strep-Tactin Superflow resin (IBA) and incubated for 90 min. The resin was washed with wash buffer (20 mM HEPES pH 7.5, 300 mM NaCl, 0.005% LMNG, 0.0005% CHS), then the protein was eluted with elution buffer (wash buffer with 10 mM d-Desthiobiotin (Sigma)). The product was further purified by size exclusion chromatography on a Superose 6 column (Cytiva) in flow buffer (20 mM HEPES pH 7.5, 100 mM NaCl, 0.005% LMNG, 0.0005% CHS). Peak fractions were pooled for nanodisc reconstitution. The sample for E2 datasets was purified the same way, except CHS was not included in the purification.

**Nanodisc reconstitution**
The plasmid for SapA expression was a gift from Salipro Biotech AB. Purification of SapA followed previously published protocols[37]. For the reconstitution of saposin nanodiscs for the PS datasets, ρ1-EM, SapA, and porcine polar brain lipid (Avanti) were mixed at a molar ratio of 1:15:150, then incubated on ice for 1 h. Bio-Beads SM-2 resin (Bio-Rad) was added into the mixture, then gently rotated overnight at 4 °C. On the next day, the supernatant was collected and further purified by gel-filtration chromatography on a Superose 6 column (Cytiva) with a buffer containing 20 mM HEPES pH 7.5 and 100 mM NaCl. Peak fractions were pooled and concentrated to ~5 mg/mL. For the E2 datasets, E2 was mixed with polar brain lipids at a 1:10 molar ratio to form the lipid mixture. The following process was the same as for the PS sample, except the E2 lipid mixture was used.

**Cryo-EM grid preparation and data collection**
Nanodisc samples were mixed with additive stock solutions in a 9:1 volume ratio, and incubated ≥30 min on ice. Stock solutions were prepared for data collection with E2 (2 mM E2, 20 mM fluorinated foscholine 8 (FFC-8), 0.5% DMSO), GABA + E2 (6 mM GABA, 2 mM E2, 20 mM FFC-8, 0.5% DMSO), PS (10 mM PS, 20 mM FFC-8, 0.5% DMSO) and GABA + PS (6 mM GABA, 10 mM PS, 20 mM FFC-8, 0.5% DMSO).

Right before application to the grid, each mixture was centrifuged to remove potential precipitation. 3 µL of the supernatant was then applied to a glow-discharged grid (R1.2/1.3 300 mesh Au grid, Quantifoil), blotted for 2 s with force 0, and plunged into liquid ethane using a Vitrobot Mark IV (Thermo Fisher Scientific).

Cryo-EM data were collected on a 300 kV Titan Krios (Thermo Fisher Scientific) electron microscope with a K3 Summit detector (Gatan) with magnification 105 k corresponding to 0.8464 Å/px using the software EPU 3.5.0 (Thermo Fisher Scientific). The total dose was ~42 e⁻/Å2, and the defocus range was −0.8 to −1.8 µm.

**Cryo-EM data processing**
Dose-fractionated images in super-resolution mode were internally gain-normalized and binned by 2 in EPU during data collection. Cryo-EM data processing was first done in RELION 3.1.4[38], including motion correction, contrast transfer function (CTF) estimation with CTFFIND 4.1[39], automatic particle picking with Topaz 0.2.5[40], particle extraction, 2D classification, 3D classification, 3D refinement, CTF refinement, and polishing. Briefly, two rounds of 2D classification were done to remove junk particles, and 3D classification (classes = 4) was used to assess structural heterogeneity. Particles from classes with protein features

were centered and re-extracted and were used for 3D refinement with C5 (E2 datasets) or C1 (PS datasets) symmetry. Multiple rounds of CtfRefine and one or two rounds of Bayesian polishing were executed to improve resolution. Shiny particles were imported into CryoSPARC v4.2.1 for further processing[41], including 3D classification in PCA mode and non-uniform refinement[42].

## Model building and refinement

Model building was started with rigid body fitting of the previously published apo (PDB ID 8OQ6) or GABA-bound (PDB ID 8OP9) structure into the density. The models were manually checked and adjusted in Coot 0.9.5[43], and ligands, water, and lipids were manually added. The resulting models were further optimized using real-space refinement in PHENIX 1.18.2[44] and validated by MolProbity[45]. Pore radius profiles were calculated using CHAP 0.9.1[46]. Structure figures were prepared using UCSF ChimeraX 1.3[47].

## Expression in oocytes and electrophysiology

mRNA encoding the ρ1-EM GABA$_A$ receptor was produced by in-vitro transcription using the mMessage mMachine T7 Ultra transcription kit (Ambion) according to the manufacturer protocol. *Xenopus laevis* oocytes (Ecocyte Bioscience) were injected with 30–50 ng mRNA and incubated 4–8 days at 13 °C in post-injection solution (10 mM HEPES pH 8.5, 88 mM NaCl, 2.4 mM NaHCO$_3$, 1 mM KCl, 0.91 mM CaCl$_2$, 0.82 mM MgSO$_4$, 0.33 mM Ca(NO$_3$)$_2$, 2 mM sodium pyruvate, 0.5 mM theophylline, 0.1 mM gentamicin, 17 mM streptomycin, 10,000 u/L penicillin) prior to two-electrode voltage clamp (TEVC) measurements. Mutagenesis was performed by methods analogous to QuikChange cloning, and the sequence was verified across the entire coding length of the gene.

For TEVC recordings, glass electrodes were pulled and filled with 3 M KCl to give a resistance of 0.5–1.5 MΩ and used to clamp the membrane potential of injected oocytes at −60 mV with an OC-725C voltage clamp (Warner Instruments). Oocytes were maintained under continuous perfusion with Ringer's solution (123 mM NaCl, 10 mM HEPES, 2 mM KCl, 2 mM MgSO$_4$, 2 mM CaCl$_2$, pH 7.5) at a flow rate of around 1.5 mL/min. Buffer exchange was accomplished by manually switching the inlet of the perfusion system to the appropriate buffer. Currents were digitized at a sampling rate of 2 kHz and lowpass filtered at 10 Hz with an Axon CNS 1440 A Digidata system controlled by pCLAMP 10 (Molecular Devices).

GABA dose-response curves in the presence and absence of steroids were measured using a 90 s co-application of 30 μM E2 or 100 μM PS during a 3.5–5.5 min pulse of GABA as shown in Fig. 1b. Each oocyte was recorded across the full GABA concentration range of 0.5–16 μM for ρ1-EM or 2–32 μM for the F283Y/F284Y mutant. Currents at the end of the initial GABA only (prior to steroid application) and GABA + steroid pulses were normalized to the maximum current measured from each oocyte. Concentration-response curves were fit using least squares regression considering each Y-value from separate oocytes as individual points using Prism with uncertainty in fitting parameters output as a 95% confidence interval.

Voltage-dependent block experiments were performed similarly to GABA and PS block experiments, with a few modifications. The holding potential for the voltage-dependent block was −80 mV, and automated voltage ramps from −80 mV to 30 mV were performed over 4 s in Ringer's solution only, upon saturation of the 1 μM GABA response, and upon saturation of the 1 μM GABA and 100 μM PS (or 0.5 μM PTX) response. Current elicited in the absence of GABA and PS was subtracted from other responses to remove contributions of leak, capacitive, and endogenous currents for each oocyte. The Woodhull model[32] was used to evaluate the fraction of the electric field traversed by the charge of PS using the following equation and fit in Prism:

$$p = K_D{}^{0mV}/(K_D{}^{0mV} + [PS]^* e^{(-n^* F^* \delta^* V/(R^* T))}) \qquad (1)$$

where $p$ is the fraction of receptors not blocked, $K_D{}^{0mV}$ is the dissociation constant at 0 mV, $[PS]$ is the concentration of PS used in the experiments (100 μM), $n$ is the charge, $F$ is Faraday's constant, $R$ is the gas constant, $T$ is temperature, $V$ is the membrane potential, and $\delta$ is the fraction of the field traversed. Voltage dependence of PTX was also fit to a Woodhull model to assess the contribution of the state-dependent block to the measured $\delta$ value. The charge was assigned as +1 due to preference of block in the resting state which is favored at negative potentials according to the rectification behavior in the control conditions. The $\delta$ value for PTX was best fit at 0.071 with a 95% confidence interval of 0.053–0.089.

## Molecular dynamics simulations

Detailed dataset-specific information can be found in Table 2. Atomic coordinates for the ρ1-EM determined by cryo-EM with different neurosteroid poses were used as starting models for MD simulations. Each subunit was split into two chains for simulation, due to the disconnection between the M3-M4 loop in the structure. The simulation systems were set up in CHARMM-GUI[48]. The protein was embedded into a lipid mixture mimicking brain lipid compositions[49], with the outer leaflet containing 152 1-palmitoyl-2-oleoyl-sn-glycero-3-phosphocholine (POPC), 14 1-palmitoyl-2-oleoyl-sn-glycero-3-phosphoethanolamine (POPE), 38 cholesterol and 15 sphingomyelin molecules, and the inner leaflet containing 34 POPC, 110 POPE, 26 1-palmitoyl-2-oleoyl-sn-glycero-3-phospho-L-serine (POPS), 32 cholesterol, 5 sphingomyelin and 18 phosphatidylinositol 4,5-bisphosphate (PIP2) molecules. The protein-lipid complex was subsequently solvated with TIP3P water and 150 mM NaCl. The CHARMM36m forcefield[50] was used to describe the proteins. Parameters for the neurosteroid were reproduced from previous work[10], in which they were generated by CGenFF[51] in CHARMM-GUI[48].

Simulations were performed using GROMACS 2021.5[52] with temperature coupled to 300 K using the velocity-rescaling thermostat[53] and pressure of 1 atm using a Parrinello-Rahman barostat[54]. The LINCS algorithm was used to constrain hydrogen-bond lengths[55], and the particle mesh Ewald method[56] was used to calculate long-range electrostatic interactions. The systems were energy minimized and then equilibrated for 20 ns, with the position restraints on the protein, and neurosteroids were gradually released. Four replicates each of 300–400 ns were simulated as final unrestrained production runs.

Before analysis, MD simulation trajectories were aligned on the Cα atoms of M2 helices by MDAnalysis[57]. Root mean square deviations (RMSD) and pore axis movement of ligands were calculated in VMD 1.9.3[58] and visualized with Matplotlib[59].

The potential of mean force (PMF) for PS permeating the pore was calculated using the accelerated weight histogram (AWH) method[60], following the protocol outlined in the GROMACS manual (https://tutorials.gromacs.org/docs/awh-tutorial.html). Relevant parameters added to the.mdp file are listed below; for access to the complete inputs, see Data Availability:

```
Awh = yes; AWH on.
Awh-nbias = 1; One bias, could have multiple.
Awh1-ndim = 1; Dimensionality of the RC
Awh1-dim1-coord-index = 1; Map RC dimension to pull the
coordinate index
Awh1-dim1-start  = − 9.1275; Sampling interval min
value (nm)
Awh1-dim1-end = 9.1275;  Sampling  interval  max
value (nm)
Awh1-dim1-force-constant = 128000; Force const of the
harmonic pot. (kJ/(mol*nm^2))
Awh1-dim1-diffusion = 5e-5; Estimate of the diffusion
(nm^2/ps)
```

The simulation was run for 650 ns with 4 walkers sharing biased data and contributing to the same target distribution. The Cα atoms of

the protein were restrained to preserve the channel in a state corresponding to experimental conditions. To prevent the neurosteroid from flipping during simulation, a flat-bottom potential of radius 8 Å was added for its upper- and lower-most atoms. Convergence was checked by monitoring the exit time of AWH from initial to final stages (580 and 560 ns for closed and desensitized simulations, respectively), and the stability of PMFs at timepoints in the final stage (Supplementary Fig. 8).

### Reporting summary

Further information on research design is available in the Nature Portfolio Reporting Summary linked to this article.

## Data availability

The cryo-EM maps and the corresponding atomic coordinates have been deposited in the Electron Microscopy Data Bank (EMDB) and the Protein Data Bank (PDB) for E2 (EMD-19167 [https://www.ebi.ac.uk/pdbe/entry/emdb/EMD-19167], PDB-8RH4 [https://doi.org/10.2210/pdb8RH4/pdb]), GABA + E2 primed state (EMD-19171 [https://www.ebi.ac.uk/pdbe/entry/emdb/EMD-19171], PDB-8RH7 [https://doi.org/10.2210/pdb8RH7/pdb]), GABA + E2 desensitized state (EMD-19172 [https://www.ebi.ac.uk/pdbe/entry/emdb/EMD-19172], PDB-8RH8 [https://doi.org/10.2210/pdb8RH8/pdb]), PS (EMD-19173 [https://www.ebi.ac.uk/pdbe/entry/emdb/EMD-19173], PDB-8RH9 [https://doi.org/10.2210/pdb8RH9/pdb]), GABA + PS (EMD-19175 [https://www.ebi.ac.uk/pdbe/entry/emdb/EMD-19175], PDB-8RHG [https://doi.org/10.2210/pdb8RHG/pdb]). MD simulation trajectories and parameter files are available in Zenodo (10406748 [https://doi.org/10.5281/zenodo.10406748]). Source data are provided in this paper.

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

## Acknowledgements

We thank professors Ryan E Hibbs, Alex Evers and the members of Molecular Biophysics Stockholm for feedback on the project and manuscript, and staff at the Cryo-EM Swedish National Facility for data collection support. Cryo-EM data were collected at the Facility funded by the Knut and Alice Wallenberg, Family Erling Persson and Kempe Foundations, SciLifeLab, and Stockholm University. MD simulations were performed using the computing facilities of the Swedish National Infrastructure for Computing (SNIC 2022/3-40, E.L.), and supported by BioExcel (EuroHPC grant no. 101093290, E.L.). J.C. was supported by an EMBO Postdoctoral Fellowship and C.F. by grant FV-5.1.2-0523-19 from Stockholm University; R.J.H. and E.L. acknowledge grants from the Swedish Research Council (2019-02433, 2021-05806) and Swedish e-Science Research Center.

## Author contributions

C.F. and J.C. performed the biochemistry, cryo-EM sample preparation, and data processing. C.F. performed model building, refinement, structural analysis, and MD simulations. J.C. performed electrophysiology. R.J.H. and E.L. supervised the project. All authors contributed to the manuscript writing and revision.

## Funding

## Competing interests

The authors declare no competing interests.
