## [Peer Review File · Nature Communications]

Divergent mechanisms of steroid inhibition in the human $\rho 1$ GABA_A receptorREVIEWER COMMENTS

Reviewer #1 (Remarks to the Author):

Fan et al. describe the results of a cryo-EM study investigating the sites and mechanisms of inhibition of the $\rho 1$ GABA-A receptor by the steroids beta-estradiol and pregnenolone sulfate. The main findings are that the steroids bind to distinct sites and act through different mechanisms. The results are of much interest to the GABA-A receptor field. I have a number of comments and criticism but all are minor and can be resolved with some additional analysis and rewriting.

1. The manuscript should make clear in Introduction that prior studies, employing mutational analysis (Li et al., Ref. 13), voltage-clamp fluorometry (Eaton, Ref. 28), and functional modeling (Germann et al., 2020; PMID: 32675382), have shown that bE2 and sulfated steroids bind to distinct sites and act through different mechanisms. The present study then goes on to show this using another, more direct (at least for sites) approach.

2. Voltage-sensitivity of PS-effect is very interesting. Please use the Woodhull model to calculate and report the fraction of the membrane field that a charge must traverse.

3. Various issues:

- l. 36, perhaps call them anion-permeable channels.
- l. 52, $\rho 3$ is also found in cerebellum as per Ref 5.
- l. 66 and elsewhere, "canonical receptor" is awkward.
- l. 76, the studies reporting the effect of the TM2-V2'S mutation on inhibition by PS (perhaps more appropriate to cite PMID: 11313438 that came out six years before the cited Ref. 16) did not propose that PS binds in the pore. The studies in fact proposed that the 2' residue is involved indirectly.
- l. 96, 97, capitalize M (molar).
- p. 4, was the F283A+F284A the only mutant tested functionally? If others were done, please report them as well even if they gave conflicting evidence.
- Figure 1H, I would recommend against illustrating the findings as inhibition ranging from 0 to 1. Perhaps use normalized current, ranging from 0 to 1, that indicates the current remaining, i.e., points at the high end would indicate more current and less inhibition.
- Figure 2A, what is relative P_o and how was it estimated?
- l. 210, it is unclear if the inhibitory mechanism can be called "block of an open pore" (which I interpret as open channel block and is defined by linear relationship between onward rate that saturates at zero current). The partial reversal of block during continued presence of PS (Fig. 3B) has also been observed in the $\alpha 1\beta 2\gamma 2L$ receptor (Pierce et al., Ref. 29) who interpreted this as redistribution of receptors among blocked, active, and desensitized (or in the authors' case, likely, resting) states.
- l. 255, perhaps use efficacious rather than potent (afterall this was done at 100 μM steroid).
- l. 255, a thing to consider here is that P_o of the response to GABA may also change with voltage, so some of the change in PS-effect may be due to change in control response. Perhaps just point this out if deemed appropriate.
- l. 267, apparent GABA affinity (more precisely, the EC50) increasing is just PS selectively

blocking high GABA currents and cutting off or imposing earlier saturation of the curve.

- l. 309, can your approach be used to calculate the predicted conductance of the resting receptor?
- l. 332, should this be Ref. 9 rather than 30?
- l. 370, the Eisenman study (Ref. 33) looked at pregnenolone sulfate, not pregnanolone sulfate.

Reviewer #2 (Remarks to the Author):

The manuscript by Fan et al describes mechanistic studies of steroid inhibition in the human $\rho 1$ GABA_A receptor. By combining several new cryo-EM structures with electrophysiology and molecular dynamics simulations, the authors characterize binding sites and negative modulation mechanisms of β -estradiol and pregnenolone sulfate at the human $\rho 1$ GABA_A receptor.

I thought that the manuscript was very insightful and well written overall. I have several relatively minor points:

- 1) Page 11, Lines 278-279. "PS in this structure was more stable with its sulfate group down rather than up in MD simulations, likely due to the 9' constriction precluding sulfate occupancy". I am not clear what this finding means for the functional mechanism. Could the authors elaborate on this?
- 2) How reliable were the force-field parameters for neurosteroids? Can the authors provide what was the range of penalty scores CGenFF outputted?
- 3) Accelerated Weight Histogram (AWH) approach for the free energy calculations is interesting but not very commonly used. Could the authors provide more details about this approach in Methods (or Supplemental methods)? I would imagine that, as in other related biased sampling methods, AWH approach must have some set of parameters that needs to be tuned and validated so that sufficient sampling is achieved. How was this done?

Reviewer #3 (Remarks to the Author):

The manuscript by Fan and colleagues investigates the mechanisms of inhibition of the $\rho 1$ GABA_A receptor by two steroids β -estradiol (E2) and pregnenolone(PS) by a combination of electrophysiology, single particle cryo-EM, and molecular dynamics simulations. E2 binds at the interface between the extracellular and transmembrane domain and without GABA almost does not affect the conformation of the receptor; upon the addition of GABA, E2 suppresses the conformational changes related to the receptor activation. PS blocks the pore of the receptor in its active state. Surprisingly, PS was also found in the pore of the receptor in the absence of the ligand.

The manuscript is well-designed and clearly written, and provides detailed insights into the inhibition of $\rho 1$ GABA_A by two steroids. While the binding sites are not novel for the GABA_A receptors in general, the details are sufficiently important for the wide audience of Nature Communications.

I have two questions which should be clarified during the revision process.

The authors assign the cryo-EM structure with E2 and GABA to the liganded pre-open state, however, the measurements of the currents showed only a minor reduction of P_o , therefore an open state should exist in a fraction of the particles of the cryo-EM dataset. For me, it is not clear if the pore is not open due to the presence of E2, or due to some other reason. From the electrophysiological measurements, I would expect that the ratio of open/closed receptors should reduce with the increased concentration of E2, however, at this moment all the receptors are closed.

This needs further examination. Ideally, an open state structure should be observed experimentally in the absence and presence of E2, however, this may come with experimental challenges. Alternatively, perhaps the authors could provide more insights into the fine details of inhibition by E2 with MD simulations.

As the authors point out - in the absence of GABA, it's hard to imagine how the PS gets inside the pore. The observation is clearly real and I think that it needs to be discussed. The authors write "The structure with PS suggests that transient rearrangements in the course of cryo-EM sample preparation, on the timescale of more than 30 minutes ligand incubation, allow the steroid to bind in the 316 resting-like state" in the results section, but they don't suggest a mechanism, can it be just diffusion? It would be very interesting to expand on this in the discussion part.

Minor: I think that the pore radius measurements from Figure S5 would be useful in the main text figures.

Erik Lindahl
Professor of Biophysics

Thank you for the thoughtful consideration of our work, "Divergent mechanisms of steroid inhibition in the human $\rho 1$ GABA_A receptor" (NCOMMS-24-22253-T). As demonstrated in the attached revision and outlined below, we believe we have responded to all concerns. In this response, "remarks to the author" are in gray italics (renumbered for clarity), our responses are indented in black and quotes from the revised manuscript are in red. Line numbers refer to the revision with figures in-line and additions tracked.

Reviewer #1 (Remarks to the Author)

Fan et al. describe the results of a cryo-EM study investigating the sites and mechanisms of inhibition of the $\rho 1$ GABA-A receptor by the steroids beta-estradiol and pregnenolone sulfate. The main findings are that the steroids bind to distinct sites and act through different mechanisms. The results are of much interest to the GABA-A receptor field. I have a number of comments and criticism but all are minor and can be resolved with some additional analysis and rewriting.

RIC1. The manuscript should make clear in Introduction that prior studies, employing mutational analysis (Li et al., Ref. 13), voltage-clamp fluorometry (Eaton, Ref. 28), and functional modeling (Germann et al., 2020; PMID: 32675382), have shown that bE2 and sulfated steroids bind to distinct sites and act through different mechanisms. The present study then goes on to show this using another, more direct (at least for sites) approach.

Thanks for this comment; we have added these studies to the updated introduction at line 85:

“Studies employing mutational analysis¹³, voltage-clamp fluorometry²⁶, and functional modeling²⁷ have shown that E2 and sulfated steroids bind to distinct sites and act through different mechanisms, though their respective details remain to be characterized.”

RIC2. Voltage-sensitivity of PS-effect is very interesting. Please use the Woodhull model to calculate and report the fraction of the membrane field that a charge must traverse.

Good idea. We have fitted the voltage dependence to the Woodhull model; see line 298 in results:

“We estimated the fraction of the electric field traversed by the charged moiety of PS upon block by fitting the voltage-dependence of inhibition to the Woodhull model³². This model relies on two free parameters, the fraction of the electric field traversed (δ) and the affinity of the blocker to the pore at 0 mV (K_{D^0mV}). Our data for PS block in $\rho 1$ -EM fits well to the Woodhull model with $\delta=0.705$ and $K_{D^0mV}=36.3 \mu M$, indicating the charge traverses $\sim 71\%$ of the electric field of the pore.”

This analysis is also described at line 582 (methods), and developed further in response to RIC3k.

RIC3. Various issues:

RIC3a. - l. 36, perhaps call them anion-permeable channels.

We have now edited this term at line 31:

“The neurotransmitter-gated γ -aminobutyric acid-A (GABA_A) receptors are anion-permeable pentameric ligand-gated ion channels expressed throughout the nervous system and other tissues.”

RIC3b. - l. 52, rho3 is also found in cerebellum as per Ref 5.

We regret overlooking this and have revised this sentence and updated the citations at line 48:

Department of Biochemistry and Biophysics

Stockholm University
Science for Life Laboratory
Box 1031
SE-171 21 Solna, Sweden

Visiting address:
Science for Life Laboratory
Tomtebodavägen 23A, Solna

Phone: +46-734618050
Cell: +46-734618050
Mail: erik.lindahl@scilifelab.se

Erik Lindahl
Professor of Biophysics

“Of the three ρ GABA_A-receptor subtypes found in mammals, $\rho 1$ is located predominantly in the retina; $\rho 2$ is more widely distributed in the brain, including the cerebellum, thalamus and frontal cortices; and $\rho 3$ is found in the hippocampus and cerebellum³⁻⁵.”

RIC3c.- l. 66 and elsewhere, "canonical receptor" is awkward.

We have replaced all 14 instances of this term with the more explicit “classical synaptic $\alpha 1\beta 2\gamma 2$ GABA_A-receptor subtype” (or variations thereon), for example at line 63:

“...allopregnanolone, which is synthesized from progesterone locally in the brain, was recently resolved by cryo-EM at this site between $\beta 1$ and $\alpha 1$ subunits in classical synaptic $\alpha 1\beta 2\gamma 2$ GABA_A receptors^{10,11}.”

RIC3d.- l. 76, the studies reporting the effect of the TM2-V2'S mutation on inhibition by PS (perhaps more appropriate to cite PMID: 11313438 that came out six years before the cited Ref. 16) did not propose that PS binds in the pore. The studies in fact proposed that the 2' residue is involved indirectly.

Appreciated. We have revised the relevant sentence and citation at line 75 of the introduction:

The specific site(s) and mechanism of PS inhibition are unclear, though physiological, biochemical, and recent structural evidence support a role for pore-facing residues in classical synaptic GABA_A receptors^{4,10,16-18}.

RIC3e.- l. 96, 97, capitalize M (molar).

These typos have been corrected beginning at line 102.

RIC3f.- p. 4, was the F283A+F284A the only mutant tested functionally? If others were done, please report them as well even if they gave conflicting evidence.

Although we expect it will be informative to pursue further explore of the E2 site by mutagenesis, the only E2 contacts we have explored thus far are F283/F284. Based on past experience in our group and others, we anticipated that only conservative substitutions in this transmembrane region would be tolerated. Given the dramatic effect of tyrosine substitutions at these positions, we prioritized sharing these results promptly, with hope of comparing other residues in and beyond the E2 site in future work.

RIC3g.- Figure 1H, I would recommend against illustrating the findings as inhibition ranging from 0 to 1. Perhaps use normalized current, ranging from 0 to 1, that indicates the current remaining, i.e., points at the high end would indicate more current and less inhibition.

With thanks for this suggestion, we have updated Figure 1H as requested.

RIC3h.- Figure 2A, what is relative P_o and how was it estimated?

Thanks for spotting this ambiguity. The more accurate description for the y-axis is I/I_{Max} ; we have adjusted the axis title in Figure 2E accordingly.

RIC3i.- l. 210, it is unclear if the inhibitory mechanism can be called "block of an open pore" (which I interpret as open channel block and is defined by linear relationship between onward rate that saturates at zero current). The partial reversal of block during continued presence of PS (Fig. 3B) has also been observed in the $\alpha 1\beta 2\gamma 2L$ receptor (Pierce et al., Ref. 29) who interpreted this as redistribution of receptors among blocked, active, and desensitized (or in the authors' case, likely, resting) states.

We agree that “block of an open pore” is not what is suggested by our structures and have adapted the subsection title paragraph text to remove this phrase and more explicitly cite the model of Pierce et al. This issue is further discussed in response to RIC31 below. The updated text can be found at line 234:

Erik Lindahl
Professor of Biophysics

“PS occludes the GABA-bound state

To explore alternative mechanisms of $\rho 1$ inhibition, we then characterized the 3 β -sulfated neurosteroid PS (Fig.3a), a negative modulator of this and several other GABA_A-receptor subtypes^{15,31}. Similar to previous reports¹³, 100 μ M PS reduced $\rho 1$ -EM currents by roughly 35%, though there was a small recovery of the inhibited current induring the continued presence of PS (Fig.3b). This partial reversal of inhibition during PS exposure has also been observed in classical synaptic GABA_A receptors, attributed to a redistribution amongst functional states upon PS binding³¹.”

RIC3j.- l. 255, perhaps use efficacious rather than potent (afterall this was done at 100 uM steroid).

Thanks for catching this; the updated text can be found at line 295:

“PS inhibition was more efficacious at more positive potentials (Fig.3h, Supplementary Fig.7h), as expected for a negatively charged blocker.”

RIC3k.- l. 255, a thing to consider here is that Po of the response to GABA may also change with voltage, so some of the change in PS-effect may be due to change in control response. Perhaps just point this out if deemed appropriate.

Appreciated; we have updated the text to acknowledge the rectification behavior in control conditions and the potential implications for measurement of the voltage-dependence of the pore block. Line 303:

“Some increase in apparent block at positive potentials may actually reflect increased channel activity with increasing voltage that also favors PS block, resulting in a potential overestimate in δ ³³. Still, this error is likely to be small, given that PTX—which shows an inverted activity-dependent block relative to PS—is nearly voltage-independent (Supplementary Fig.7h-j). Comparison to PTX profiles suggests the activity dependence of block contributes 0.05 to 0.09 of the apparent δ value.”

RIC3l.- l. 267, apparent GABA affinity (more precisely, the EC50) increasing is just PS selectively blocking high GABA currents and cutting off or imposing earlier saturation of the curve.

Thanks for this thoughtful proposal. We agree part of the reason for increased apparent GABA affinity is PS selectively blocking high GABA currents. However, we respectfully submit that Figure 1H suggests another mechanism is at play. The increase in current amplitude upon washout of PS beyond the steady state level of GABA alone indicates that channel activity is higher after PS treatment than before. A likely explanation of this is that the receptor GABA affinity increases during PS treatment; upon washout, PS unbinding is faster than re-equilibration of the GABA-bound fraction of receptors. This model would also help explain the partial reversal of channel inhibition during sustained presence of PS (see RIC3i above). Building on that response, we have added text clarifying our model at line 243:

“Unlike E2, PS wash-out was associated with a transient recovery current ~30% larger than steady-state GABA activation prior to treatment (Figure 3B). This behavior is expected for a pore blocker that preferentially binds to the activated (open or desensitized) state of the pore. Consistent with this model, current recovery upon PS washout was not observed at high concentrations of GABA, where channels are fully activated prior to PS treatment (Supplementary Fig.7f).”

RIC3m.- l. 309, can your approach be used to calculate the predicted conductance of the resting receptor?

The potential of mean force (PMF) calculation we used gives the free energy surface along the chosen collective variable, in this case, the pathway for PS along the ion channel pore. It can provide information regarding the free energy barrier for an ion or ligand to transit the pore, which is related to conductance, but to our knowledge is not a straightforward method to directly predict ion conductance values; typically, more extensive computational electrophysiology simulations would be required. In relative terms, our data do indicate a ~100-kJ/mol barrier to PS translocation past the 9' gate in the resting-like

Erik Lindahl
Professor of Biophysics

state; it can be inferred that chloride ions would also encounter a high barrier, rendering conductance negligible—as also indicated by the pore radius, smaller than a hydrated chloride ion. We have added text to the discussion to clarify this point at line 453, also detailed in response to R3C2:

“Although PS binding in the resting-like state is evidently possible in the context of prolonged incubation for cryo-EM, it would be disfavored by the permeation barrier at the 9' gate. This barrier was estimated at 100 kJ/mol by our PMF calculations: although these measurements do not directly report conductance, the central hydrophobic barrier is expected to preclude inward transit of PS as well as chloride ions in a physiological setting.”

R1C3n.- l. 332, should this be Ref. 9 rather than 30?

It should indeed. Sorry - fixed now!

R1C3o.- l. 370, the Eisenman study (Ref. 33) looked at pregnenolone sulfate, not pregnanolone sulfate.

Thanks for this comment. Although the focus of the Eisenman study is on pregnenolone sulfate, these authors do compare the voltage-dependence of inhibition for pregnanolone sulfate (3 α 5 β PS) and show this comparison in Figure 2C noting the contrasting behavior of the two sulfated steroids. We have therefore respectfully retained our original description of this result at line 439:

“Interestingly, inhibition by the related compound pregnanolone sulfate was previously shown to be voltage-dependent at α 1 β 2 γ 2 GABA_A receptors, but voltage-independent at ρ 1^{13,33}.”

Reviewer #2 (Remarks to the Author)

The manuscript by Fan et al describes mechanistic studies of steroid inhibition in the human ρ 1 GABA_A receptor. By combining several new cryo-EM structures with electrophysiology and molecular dynamics simulations, the authors characterize binding sites and negative modulation mechanisms of β -estradiol and pregnenolone sulfate at the human ρ 1 GABA_A receptor. I thought that the manuscript was very insightful and well written overall. I have several relatively minor points:

R2C1. Page 11, Lines 278-279. “PS in this structure was more stable with its sulfate group down rather than up in MD simulations, likely due to the 9' constriction precluding sulfate occupancy”. I am not clear what this finding means for the functional mechanism. Could the authors elaborate on this?

We regret that this observation was not sufficiently clear. Likely due to the relative constriction of the pore in resting-like vs. desensitized states, the orientation and pose of PS indeed differed in the absence and presence of GABA. When physiologically relevant, such differences can have implications for e.g. mechanistic modeling and pharmaceutical development. However, given the barrier to PS permeation across the resting-like 9' gate, and lack of evidence for resting-state PS stabilization in the presence of GABA (in contrast to PTX or E2), binding of PS in the resting-like state is not likely to contribute substantially in a functional setting. Transient expansion of one or more permeation gates may enable PS diffusion on the >30-min timescale of cryo-EM sample preparation, but the state thus captured may be largely an artefact of the conditions. We therefore refrain from extensive speculation as to the relevance of PS orientation in this state, while clarifying the observation at line 352 (see also response to R3C2):

“The structure with PS suggests that transient rearrangements in the course of cryo-EM sample preparation, on the timescale of more than 30 minutes ligand incubation, allow the steroid to bind in the resting-like state, with apparent alterations in orientation and pose to accommodate the constricted pore. However, structures in the presence of GABA show that a closed pore is not preferentially stabilized by PS as it is by PTX or E2, and the complex with PS and GABA appears to be the more relevant model for ρ 1 functional inhibition.”

Erik Lindahl
Professor of Biophysics

R2C2. How reliable were the force-field parameters for neurosteroids? Can the authors provide what was the range of penalty scores CGenFF outputted?

Parameters or PS were taken from previously work on the synaptic-subtype GABA_A receptor (Ref. 10). Ligand RMSD was <3 Å over hundreds of nanoseconds in its more favored pose, supporting the potential for these parameters to model stable interactions. Penalty scores for most parameters generated for PS were ≤65, aside from four dihedrals with higher penalties. For transparency, the parameter files have been uploaded to the reviewer Drive, and the corresponding 3D structure is depicted below from two angles; no unreasonable features were evident. We have also clarified the methods at line 611:

“Parameters for the neurosteroids were reproduced from previous work¹⁰, in which they were generated by CGenFF⁵¹ in CHARMM-GUI⁴⁸.”

R2C3. Accelerated Weight Histogram (AWH) approach for the free energy calculations is interesting but not very commonly used. Could the authors provide more details about this approach in Methods (or Supplemental methods)? I would imagine that, as in other related biased sampling methods, AWH approach must have some set of parameters that needs to be tuned and validated so that sufficient sampling is achieved. How was this done?

Although relatively recent (Ref 60, 2018), AWH calculations have been applied to a range of systems including multiple ion channels. As detailed in the regularly updated GROMACS manual (<https://manual.gromacs.org/current/reference-manual/special/awh.html>), it applies a time-dependent potential along a reaction coordinate, which is tuned to flatten barriers to improve sampling. We chose this approach because of the robustness with respect to the parameterization, as demonstrated for example by recent work comparing AWH with umbrella sampling (<https://chemrxiv.org/engage/chemrxiv/article-details/664e661891aefa6ce1bbc376>). Briefly, AWH is initialized with an estimate of the free energy, which is then updated at regular intervals using simulations data. The only parameter to be tuned is the initial update size (controlled by the `awh1-dim1-diffusion` parameter in the GROMACS mdp file), which must be large enough that the bias fluctuations efficiently promote escaping free energy minima. We have added a more detailed description of relevant parameters at line 624 in methods:

The potential of mean force (PMF) for PS permeating the pore was calculated using the accelerated weight histogram (AWH) method⁶⁰, following the protocol outlined in the GROMACS manual (<https://tutorials.gromacs.org/docs/awh-tutorial.html>). Relevant parameters added to the .mdp file are listed below; for access to the complete inputs, see Data Availability:

```
awh = yes ; AWH on.
awh-nbias = 1 ; One bias, could have multiple.
awh1-ndim = 1 ; Dimensionality of the RC
awh1-dim1-coord-index = 1 ; Map RC dimension to pull coordinate index
awh1-dim1-start = -9.1275 ; Sampling interval min value (nm)
awh1-dim1-end = 9.1275 ; Sampling interval max value (nm)
awh1-dim1-force-constant = 128000 ; Force const of harmonic pot. (kJ/(mol*nm^2))
awh1-dim1-diffusion = 5e-5 ; Estimate of the diffusion (nm^2/ps)
```

Erik Lindahl
Professor of Biophysics

In the initial stages of AWH, fluctuations in the free energy estimate and bias are large since sampling is favored, but simulations typically converge in hundreds of nanoseconds. The exit time associated with passing the standardized threshold for convergence was 580 and 560 ns respectively for our simulations of resting-like and desensitized channels with PS. To verify convergence, the stability of the PMF across several timepoints subsequent to the exit time can also be checked. We have now added convergence plots for the PS systems to the manuscript (Fig. S8), and expanded the topic at line 641 (methods):

“Convergence was checked by monitoring the exit time of AWH from initial to final stage (580 and 560 ns for closed and desensitized state simulations respectively), and the stability of PMFs at timepoints in the final stage (Supplementary Fig.8).”

Reviewer #3 (Remarks to the Author):

The manuscript by Fan and colleagues investigates the mechanisms of inhibition of the $\rho 1$ GABA_A receptor by two steroids β -estradiol (E2) and pregnenolone(PS) by a combination of electrophysiology, single particle cryo-EM, and molecular dynamics simulations. E2 binds at the interface between the extracellular and transmembrane domain and without GABA almost does not affect the conformation of the receptor; upon the addition of GABA, E2 suppresses the conformational changes related to the receptor activation. PS blocks the pore of the receptor in its active state. Surprisingly, PS was also found in the pore of the receptor in the absence of the ligand.

The manuscript is well-designed and clearly written, and provides detailed insights into the inhibition of $\rho 1$ GABA_A by two steroids. While the binding sites are not novel for the GABA_A receptors in general, the details are sufficiently important for the wide audience of Nature Communications.

I have two questions which should be clarified during the revision process.

R3C1. The authors assign the cryo-EM structure with E2 and GABA to the liganded pre-open state, however, the measurements of the currents showed only a minor reduction of P_o , therefore an open state should exist in a fraction of the particles of the cryo-EM dataset. For me, it is not clear if the pore is not open due to the presence of E2, or due to some other reason. From the electrophysiological measurements, I would expect that the ratio of open/closed receptors should reduce with the increased concentration of E2, however, at this moment all the receptors are closed.

This needs further examination. Ideally, an open state structure should be observed experimentally in the absence and presence of E2, however, this may come with experimental challenges. Alternatively, perhaps the authors could provide more insights into the fine details of inhibition by E2 with MD simulations.

Thanks for this thoughtful comment. It is indeed interesting that relatively little change in current is observed in our electrophysiology experiments at 16 μ M GABA when 30 μ M E2 is applied given that roughly half of the particles remain in a resting-like state in the cryo-EM dataset in the presence of GABA and E2. However, the differences in E2 concentrations between electrophysiology and cryo-EM experiments makes direct comparison difficult. The solubility limit of E2, even in the presence of DMSO, restricts electrophysiology to < 30 μ M E2, whereas much higher concentrations can be achieved in cryo-EM for two reasons: First, E2 was added to the lipid mixture used for nanodisc reconstitution so that -10% of total lipids are E2. Second, the detergent FFC-8 added to prevent orientation bias also facilitates solubilization of E2, enabling a final concentration of -200 μ M in the cryo-EM sample. We would expect a more substantial reduction in current at saturating GABA concentrations if a similar E2 concentration could be achieved in electrophysiology experiments. We now discuss this discrepancy at line 200:

“On the other hand, given the limited effect of E2 with saturating GABA in electrophysiology experiments, it may seem surprising that E2 promotes/displays such a substantial population in the primed state by cryo-EM, even under high-GABA conditions that would produce a single desensitized state without E2⁸. However, concentrations of E2 applied in the cryo-EM samples are roughly an order of magnitude higher than concentrations used in electrophysiology due to the improved solubility of E2 in

Erik Lindahl
Professor of Biophysics

the presence of lipids and detergents used in grid preparation. Thus, it is difficult to directly assess the functional effect of E2 at cryo-EM concentrations.“

Regarding the idea to explore E2 inhibition by simulations, although we considered several possibilities, classical MD does not sample timescales to meaningfully inform shifts in the $\rho 1$ conformational landscape upon ligand binding—particularly in the absence of a definitive open structural template. As the reviewer rightly points out, it remains a substantial limitation (as for other Cys-loop receptors) that we still do not have any open state of $\rho 1$. Based on functional behavior, we would have expected a partly open population; instead, all our GABA-bound structures appear to be desensitized. The reason why remains unclear; as described in our previous work (Ref. 8), it may reflect cryo-EM conditions, such as interactions with the membrane mimetic or air-water interface, that bias open channels to desensitize. Interestingly, a previous study using both radioligand binding and electrophysiology indicated that only 10% of receptors on the cell surface conduct currents in macroscopic recordings (Chang and Weiss, 1999, PMID 10195213); thus the maximal open population in functional experiments may represent only a small fraction of receptors. We have revised the discussion of this (line 465):

“These findings may indicate that cryo-EM sample conditions favor a desensitized-like structure that does not perfectly represent the physiological ensemble³⁶, which should include one or more open structures.”

Although we cannot rule out differential interactions of open $\rho 1$, we predict that the desensitized state captured in our micrographs to be a representative proxy for E2 interactions—or lack thereof. Due to rearrangements including outward translocation of M2 and the M2–M3 loop and reorientation of R337, the E2 binding pocket does not appear to accommodate E2 in the desensitized state. Since rearrangements between GABA-bound open and desensitized states are thought to involve the intracellular end of the TMD, clashes in the E2 pocket in the desensitized state would like also be present in the open state. We have expanded the discussion of this point at line 402:

“Particularly in the absence of a definitive open state, the limited timescales of classical MD preclude definitive modeling of mechanistic effects of an allosteric ligand like E2; nonetheless, conformational changes in the desensitized state due to expansion of the upper pore are likely to also reflect the open state, and could similarly block E2 binding.”

R3C2. As the authors point out - in the absence of GABA, it's hard to imagine how the PS gets inside the pore. The observation is clearly real and I think that it needs to be discussed. The authors write “The structure with PS suggests that transient rearrangements in the course of cryo-EM sample preparation, on the timescale of more than 30 minutes ligand incubation, allow the steroid to bind in the resting-like state” in the results section, but they don't suggest a mechanism, can it be just diffusion? It would be very interesting to expand on this in the discussion part.

As discussed in response to R1C3m and R2C1, capturing PS in the resting state with both 9' and -2' gate closed is indeed interesting. The 100 kJ/mol barrier across the resting-like 9' gate, and limited permeability of negatively charged PS to enter from the intracellular side, suggest this state would not contribute substantially in a physiological setting. We make the educated guess that transient expansion of one or more gates enables PS diffusion on the >30-min timescale of cryo-EM sample preparation, possibly facilitated by slow transitions of neighboring residues or lipids not accessible on the simulation timescales. We have clarified some of this in results (lines 352, see R2C1) and discussion (lines 453, see R1C3m), but refrained from extensive speculation as to the relevance of PS binding in the resting state.

R3C3. Minor: I think that the pore radius measurements from Figure S5 would be useful in the main text figures.

Good suggestion; we have moved the pore radius profiles to Figure 2f, 2g.

Once again, thank you so much for extensive highly productive comments that helped improve our work.

REVIEWER COMMENTS

Reviewer #1 (Remarks to the Author):

The authors have addressed my original comments. For further improvement, I have a suggestion and two related requests for clarification:

- Re-word "independent oocytes". Perhaps use individual or separate.
- Provide more details on how the experiments in Fig. 2e were done. So, for GABA dose-response curve, I would think, the authors did 3.5-5.5 min applications of 0.5-20 (or is it 15?) μM GABA, separated by washouts of unspecified duration in bath. All GABA concentrations were tested in all 5 cells and for plotting normalized to the peak response to some unspecified (appears a bit less than 10 μM in the plot) GABA concentration? Do the lines in Fig. 2e show fits to pooled, averaged data, or were the data from each individual cell fitted separately and the presented EC_{50} is a mean from 5 individual fits? Please clarify and add to the Methods section. In either case there should also be some measure of uncertainty provided for the EC_{50} s.
- Further, how were the experiments in the presence of 30 μM E2 done? Does the I/I_{max} represent the inhibitory effect of E2 at that specific GABA concentration or is it the ratio of current in the presence of steroid to the current at saturating GABA in the same oocyte? In other words and for example, does the I/I_{max} of ~ 0.4 in the presence of $\sim 2 \mu\text{M}$ GABA + 30 μM E2 indicate that E2 inhibited the 2 μM GABA response to 40% of the 2 μM GABA response or that the 2 μM GABA + 30 μM E2 response equaled 40% of the saturating GABA-alone response in the same oocyte?

Reviewer #2 (Remarks to the Author):

The revised manuscript satisfactorily addresses all my previous comments.

Reviewer #3 (Remarks to the Author):

The authors addressed by comments in the updated version of the manuscript, I favor its publication.

Reviewer #1 (Remarks to the Author)

The authors have addressed my original comments. For further improvement, I have a suggestion and two related requests for clarification:

- Re-word "independent oocytes". Perhaps use individual or separate.

Check. We have changed all these phrases to use "individual" instead (lines 739, 758, 785), and similarly in the extended data.

- Provide more details on how the experiments in Fig. 2e were done. So, for GABA dose-response curve, I would think, the authors did 3.5-5.5 min applications of 0.5-20 (or is it 15?) μM GABA, separated by washouts of unspecified duration in bath. All GABA concentrations were tested in all 5 cells and for plotting normalized to the peak response to some unspecified (appears a bit less than 10 μM in the plot) GABA concentration? Do the lines in Fig. 2e show fits to pooled, averaged data, or were the data from each individual cell fitted separately and the presented EC50 is a mean from 5 individual fits? Please clarify and add to the Methods section. In either case there should also be some measure of uncertainty provided for the EC50s.

- Further, how were the experiments in the presence of 30 μM E2 done? Does the I/Imax represent the inhibitory effect of E2 at that specific GABA concentration or is it the ratio of current in the presence of steroid to the current at saturating GABA in the same oocyte? In other words and for example, does the I/Imax of ~0.4 in the presence of ~2 μM GABA + 30 μM E2 indicate that E2 inhibited the 2 μM GABA response to 40% of the 2 μM GABA response or that the 2 μM GABA + 30 μM E2 response equaled 40% of the saturating GABA-alone response in the same oocyte?

To address both of these, we have expanded the methods section starting at line 481:

"Each oocyte was recorded across the full GABA concentration range of 0.5 to 16 μM for $\rho 1\text{-EM}$ or 2 to 32 μM for the F283Y/F284Y mutant. Currents at the end of the initial GABA only (prior to steroid application) and GABA+steroid pulses were normalized to the maximum current measured from each oocyte. Concentration-response curves were fit using least squares regression considering each Y-value from separate oocytes as individual points using Prism with uncertainty in fitting parameters output as a 95% confidence interval."

We have also modified the Fig. 2e caption to clarify the uncertainty by reporting 95% confidence intervals both for GABA-only and GABA+E2 measurements:

"...(GABA alone, 95% confidence interval 1.16-1.41 μM) or 1.8 μM (GABA+E2, 95% confidence interval 1.56-2.21 μM)."

We have also updated the captions of Extended Data Fig. 5b,c and Fig. 7g similarly.

Department of Biochemistry and Biophysics

Stockholm University
Science for Life Laboratory
Box 1031
SE-171 21 Solna, Sweden

Visiting address:
Science for Life Laboratory
Tomtebodavägen 23A, Solna

Phone: +46-734618050
Cell: +46-734618050
Mail: erik.lindah@scilifelab.se